# A Comprehensive Review of Hypotheses About the Biological Function of Zearalenone, and a New Hypothesis for the Function of Resorcylic and Dihydroxyphenylacetic Macrolactones in Fungi

**DOI:** 10.3390/toxins17050226

**Published:** 2025-05-03

**Authors:** María Viñas, Petr Karlovsky

**Affiliations:** 1CIGRAS, University of Costa Rica, San Jose 2060, Costa Rica; maria.vinasmeneses@ucr.ac.cr; 2Molecular Phytopathology and Mycotoxin Research, University of Göttingen, 37077 Göttingen, Germany

**Keywords:** zearalenone, resorcylic acid lactone, dihydroxyphenylacetic acid lactone, interference competition, chemical defense, radicicol, curvularin, fusaristatin, mycoparasitism, competition

## Abstract

The special metabolite of *Fusarium* spp. zearalenone (ZEN) exerts estrogenic effects on mammals, stimulates plant growth, stimulates sexual development in fungi, and inhibits fungal growth. These activities inspired hypotheses about the biological function of ZEN. We briefly review the discovery of ZEN and its implications. The main subject of this review is a critical assessment of the hypotheses that ZEN is a fungal hormone, a plant hormone, a virulence factor, or a fungal defense metabolite. Conceptual and technical issues related to testing these hypotheses, such as inadequate analytical methods, confusion of incidental effects with biological functions, and lack of normalization, are illuminated. Based on these considerations, gene knockout experiments, and on the effects of biotic interactions on ZEN synthesis, we argue that ZEN is a defense metabolite protecting *Fusarium* spp. against mycoparasites and competitors. Similar reasoning and published data suggest that the *Fusarium* metabolite fusaristatin A fulfils the same function. Fungi produce many macrolactones of resorcylic acid (RALs) and dihydroxyphenylacetic acid (DHPLs) with properties similar to ZEN. Their widespread occurrence, antifungal activity, and further considerations prompt us to hypothesize that the fundamental function of fungal RALs and DHPLs lies in defense and interference competition.

## 1. Objective and Scope

Zearalenone (ZEN), a metabolite of plant-pathogenic fungi of the genus *Fusarium*, affects food production in two ways. As a mycotoxin, ZEN contaminating cereal grains causes hyperestrogenism in farm animals and, rarely, also in humans. Maximum limits for ZEN in food and feed have therefore been established (e.g., [1]), though the acute toxicity of ZEN is low [2] and there is no evidence of carcinogenicity in humans [3]. The vast majority of experimental studies and reviews on ZEN have been devoted to toxicology, occurrence, exposure, and prevention. We will skip these topics entirely. On the other hand, ZEN contributes to meat production because it is used to manufacture growth stimulants (anabolics) for livestock. Their development is closely connected with the discovery of ZEN and early research into its biological activities, which laid the groundwork for the initial hypothesis about the biological function of ZEN. This research will be covered in the second section, together with its unexpected societal implications.

As compared to extensive applied research on ZEN, the biological function of ZEN has received limited attention. Four hypotheses posited that ZEN is a fungal hormone, a plant hormone, a virulence factor, or a fungal defense metabolite. Critical assessment of these hypotheses in Section 5, Section 6, Section 7 and Section 8 is our main subject. All four hypotheses have been motivated by the effects of ZEN on other organisms. These effects may provide hints regarding the ecological interactions of ZEN-producing fungi, but inferring functions from effects can be misleading. Even conspicuous activities may have originated from side effects unrelated to the function maintained by natural selection. Similarly, detoxification of ZEN by organisms that are naturally exposed to ZEN may be unrelated to the biological function of ZEN, even if it underlies positive selection. A critical interpretation of experimental data used to support the hypotheses, results to the contrary, data from gene knockout experiments, and the effect of interactions with other organisms on ZEN synthesis by *F. graminearum* all lead us to argue that ZEN is a defense metabolite protecting *Fusarium* spp. from mycoparasites and competitors.

Fungi produce numerous macrolactones of resorcylic acid (RALs) and dihydroxyphenylacetic acid (DHPLs) with structures and biological activities similar to ZEN. We hypothesize that the fundamental function of these metabolites is the protection of their producers from mycoparasites and competitors. In other words, we suggest that they are agents of chemical defense and interference competition. The hypothesis is supported by antifungal effects of these metabolites, widespread occurrence of specific lactonases in fungi, considerations about the lifestyle of saprotrophs and necrotrophs, and the effect of biotic interactions on the synthesis of these metabolites. These considerations also allow us to offer a plausible explanation for recently published puzzling results on a *Fusarium* metabolite, fusaristatin A, which negatively affects the saprophytic as well as the pathogenic stage of the life cycle of *Fusarium graminearum*, yet its production has not been counterselected.

In addition to the main objectives, several related topics that have not been covered adequately in the literature, provide interesting narratives, or deserve clarification, are addressed. The first, covered in the description of early research on ZEN, is the development of a growth stimulant for animal husbandry, which triggered a trade war between the USA and the European Union lasting three decades. ZEN is the only mycotoxin that has affected international trade to such an extent. Further topics covered are the questions of whether the ratio of isomers of zearalenol in field grain corresponds to the ratio observed in plants treated with ZEN in vitro and the taxonomic affiliation of *Fusarium* spp. producing ZEN. The latter question has been controversial, with disputed claims persistently appearing in scientific journals and even in popular media. We attempt to unravel the controversy, explaining technical shortcomings underlying the incorrect claims and disclosing the likely origin of a recent incorrect assignment of certain genes in the *Fusarium verticillioides* genome to ZEN synthesis.

## 2. Discovery and Practical Use of ZEN

Both the discovery of ZEN and research into its properties were fueled by conspicuous hyperestrogenic effects observed in pig farms and by commercialization efforts triggered by these observations. Initial research into ZEN is best described by combining academic and commercial perspectives.

Since the 1920s, farmers in the USA reported swollen vulva and mammary glands in sows, in severe cases progressing to prolapses of the vagina and rectum, and inflammation of the prepuce and atrophy of the testes in male pigs. These observations were first reported in 1928 from the USA [4] and later from Australia [5] and Ireland [6]. In years with wet weather before the harvest, fertility decline due to hyperestrogenism continues to plague pig farms till today [7]. Because of its estrogenicity, ZEN is legally treated as a mycotoxin, and legal limits for the maximum content of ZEN in food and feed have been established (e.g., [1]), though the acute toxicity of ZEN is low [2] and there is no evidence of carcinogenicity of ZEN in humans [3].

In the 1950s, two research groups in the USA set out to identify the cause of alimentary hyperestrogenism in pigs. At Pardue University, Martin Stob’s lab grew fungal strains isolated from moldy feedstuff and tested the culture extracts for the ability to cause hyperestrogenic syndrome. Only extracts of *Fusarium graminearum* cultures reproduced the syndrome. The Pardue group purified the active agent, which they called “anabolic substance” in their key patent of 1965 [8]. Because the metabolite was found to be essentially insoluble in water, fungal cultures on maize grains were blended with water and filtered to remove water-soluble contaminants, and the anabolic substance was extracted into an organic solvent. Purification to homogeneity was achieved by a series of extractions from organic into aqueous phase and back under alkaline and acidic conditions, respectively, followed by a manual predecessor of countercurrent chromatography. In the patent, the inventors reported that injecting the substance stimulated weight gain in sheep (Figure 1). Interestingly, they also presented a correct structural formula without revealing the source of this information. The structure of ZEN was published [9] two years after the submission of the patent application and one year after the patent was issued.

### 2.1. Zearalenone Derivatives as Commercial Anabolics

Although the key patent on the use of ZEN as an anabolic agent was owned by Pardue University, collaboration of the Pardue group with the Commercial Solvent Corporation (CSC), located a 1.5 h drive from their campus, began much earlier. In 1962, the Pardue group and CSC jointly published the first report of estrogenic and anabolic properties of a metabolite purified from extracts of *F. graminearum* cultures [10]. (In this work, they apparently stumbled over trichothecenes: “An easily separable, water-soluble fraction, toxic for mice, is occasionally produced concurrently with the metabolite”). After R. Baldwin from the Pardue Group moved to CSC, the company took over the project. In 1975, CSC was acquired by the International Minerals and Chemical Corporation, giving rise to IMC Global.

The elucidation of the structure of ZEN was accomplished by the joint efforts of Wilbert Urry at the University of Chicago and CSC in 1966 [9]. They established the trivial name zearalenone.

The second team searching for the causal agent of estrogenic syndrome in farm animals was the Minnesota group, which consisted of Chester Mirocha, Clyde Christensen, and Glen Nelson at the University of Minnesota. They, too, tested extracts of *F. graminearum* cultures for estrogenic and anabolic activity. Bioassays on pigs and rats led them to a single active metabolite, which they designated F-2 [11]. The metabolite turned out to be identical to ZEN, but the Minnesota group continued using the acronym F-2 long after the trivial name ZEN had been established.

The anabolic effect of ZEN in lambs described in the key patent [8] was not statistically significant, but—together with other results such as those from the Minnesota group in rats [12]—was sufficient to spark a major project at CSC. Over 100 derivatives of ZEN have been synthesized and their pharmacological properties tested (reviewed by Shipchandler in [13]). The most promising derivative, in which the double bond was hydrogenated and the keto group reduced to α-ZAL (Figure 2), was marketed as Ralgro. The compound, also known as zeranol, was delivered as subcutaneous deposits for calves and sheep and became a commercial success. Apart from Ralgro, the product has also been marketed under several brand names, including Frideron, Ralabol, Ralone, Zerano, and Zeranol.

The β epimer of zearalanol was developed under the trivial names taleranol and teranol, but it has never been commercialized. To our knowledge, α-ZAL implants have been approved exclusively for beef cattle and sheep, though a single research report indicated that they might also benefit pig production [14].

The yield of ZEN according to the original protocol of the Pardue group [8] was low. CSC optimized the fermentation, purification, and conversion of ZEN to α-ZAL. The improved protocols are described in a series of patents issued between 1966 and 1976 [15,16,17,18,19,20,21]. After the patents expired, other companies started producing α-ZAL while further optimizing the process. α-ZAL was originally obtained from ZEN by hydrogenation on Raney nickel and later palladium on charcoal, which leads to a mixture of α-ZAL and β-ZAL. β-ZAL, as an undesirable side product, had to be removed. Because α-ZAL and β-ZAL are diastereomers, conversion catalyzed by a non-chiral catalyst may exhibit stereoselectivity. Indeed, using nonselective Meerwein–Ponndorf–Verley catalysts, the yield was gradually improved from 50% to 85% [22,23]. With bulky organoborane as an intermediate, which was removed before hydrogenation of the double bond, a yield of 95% α-ZAL was achieved [24]. Nearly stereospecific reduction was also achieved with the complex of LiAlH_4_ with 3-O-benzyl-1,2-O-cyclohexylidene-α-glucofuranoside [24]. The desired isomer was purified by crystallization, taking advantage of the lower solubility of α-ZAL. Isopropanol/water mixture was used as a solvent. In 1989, Italian company CRC Compagnia di Ricerca Chimica applied for a patent on an improved separation of the diastereomers of ZAL by crystallization from acetonitrile with water [25], but the application was withdrawn. The separation of α- and β-ZAL by the enzymatic hydrolysis of a mixture of 7-acyl-zearalanols has been described [26], but to our knowledge, it has not been used commercially. Apart from α-ZAL and ZAN, other derivatives such as 2-methylether-4-methoxyacetete, which was patented by Merck [27], and zearalenone glycoside, which was patented by CSC [28], were shown to possess anabolic activities. None has been commercialized.

### 2.2. High Yields of ZEN from Fermentation

Already in 1971, in a discussion after Nelson’s report [29], Mirocha revealed that their solid-state cultures of *F. graminearum* accumulated 20–30 g ZEN/kg. A yield of 17 g ZEN per liter of liquid medium was reported in a patent of IMC Chemical Group [21]. Up to 38 g ZEN/kg from solid-state fermentation were reported by the University of Minnesota [30]. Marasas et al. [31], using published data [32], calculated a yield of 50 g ZEN/kg. Kuiper-Goodman et al., in their reviews, gave 60 g ZEN/kg as the maximum level produced in culture. These are orders of magnitude more than other *Fusarium* mycotoxins such as enniatins, fumonisins, deoxynivalenol, or T-2 toxin. We speculate that high levels of ZEN in substrates colonized by *Fusarium* spp. might be related to its biological function (Section 8).

### 2.3. Laboratory Synthesis of ZEN

Soon after the structure of ZEN became known, the first total synthesis of racemic ZEN was patented by Merck [33]. The synthesis consisted of 14 steps, and the yield was low. Protecting the method by a patent appears unusual today because its use in industrial production was unlikely. We speculate that the patent was part of Merck’s strategy to gain a foothold in the promising field. The company also synthesized pharmacologically interesting derivatives of ZEN [27], and it later acquired the rights to Ralgro and became one of its major distributors. The chemists at Merck also published their synthesis in scientific journals [34,35], starting a race for synthetic routes for ZEN, which counts over two dozen methods and continues till today [36].

### 2.4. Zeranol as a Trigger of Trade War Between the European Union and the United States

Ralgro was approved in the US in 1969 for beef cattle [37], and its use rapidly spread. In the EU, all hormonal growth regulators for livestock have been banned since 1981 [38]. The ban covered the use of zeranol in animal husbandry as well as the processing and sale of meat from animals treated with zeranol or other growth stimulators. The US challenged the ban at the WTO on the grounds that it violated the General Agreement on Tariffs and Trade (GATT). While the WTO started a dispute, the EU delayed the ban.

The dispute was not settled till the end of 1988. With effect from January 1989, the EU imposed an import ban on meat from animals treated with zeranol, effectively cutting off US beef export to Europe [39], estimated at USD 202 million annually [40]. In response, the US enacted retaliatory tariffs on products from the EU [41]. The list of products has been revised every year to maximize the pressure.

The dispute at the WTO continued for two decades, with both parties presenting scientific reviews and claiming procedural errors. Till 2003, 51 complaints had been filed. In 2003, the EU replaced the ban with a provisional ban, referring to a WTO rule allowing provisional measures during ongoing risk assessments. Based on this change, the EU claimed that punitive tariffs by the US and Canada violated WTO rules [42]. The WTO panel repeatedly delayed its decision, and the United States sustained the carousel retaliation despite fears that the EU could retaliate in return [43]. In 2008, the WTO panel issued a decision, ruling that neither the EU ban nor the trade sanctions by the United States and Canada were justified. Both parties appealed. The panel reversed its decision, approving the ban and the sanctions.

In December 2008, the EU initiated direct negotiations with the US [44]. In January 2009, the Bush administration issued the last revision of the carousel retaliation shortly before the transition of power. The Obama administration put the retaliatory tariffs on hold and started direct negotiations with the EU. On 13 May 2009, the EU and the US signed a memorandum of understanding [45]. The EU committed to importing a certain amount of hormone-free beef, and the US to impose import duties on a “reduced list” of products.

In 2012, the EU increased the quota of imported hormone-free beef [46], and the US suspended the retaliatory tariffs. In the following years, the US complained that most of the increased imports by the EU were filled by Australia, Argentina, New Zealand, and other countries, while the EU insisted on a “first come, first served” system. The US took steps to reinstate retaliatory tariffs [47], which were imposed in early 2019. The EU reviewed its import quota, and both parties renegotiated. A final agreement was reached in June 2019. Phased in over a seven-year period, the EU reserved 78% of beef imports to the US, and the US waived the punitive tariffs [48]. Three decades of trade war due to a product manufactured from ZEN came to an end.

### 2.5. Zeranol as an Illicit Doping Agent

Due to its anabolic potential in humans, the World Anti-Doping Agency (WADA) classified zeranol as an illicit doping agent [49], and zeranol has been included in routine doping controls since 1995 [50].

In 1996, zeranol was detected in the urine of an athlete for the first time [50]. The consequences for the athlete are unknown. The following cases have been reported by the mass media. In 2014, Chinese hammer thrower Zhang Wenxiu tested positive for zeranol and was stripped of the gold medal. She successfully appealed. In 2016, Jamaican fitness athlete Deidre Lewis tested positive for zeranol. She was banned from competition for two years. In 2017, American runner Ajeé Wilson tested positive for zeranol. She was stripped of her 800 m national record but not banned from future competitions. In 2019, American rugby player Mahlatse Ralepelle tested positive for zeranol. He was banned for eight years. In 2022, Swiss mountain biker Mathias Flückiger tested positive for zeranol. He was provisionally suspended, challenged the decision, and was cleared of the allegation.

The reason for the variation of the penalties is that zeranol may be ingested with meat from treated animals, and the human body can convert ZEN ingested with food into zeranol. This makes the distinction between misuse and unintentional exposure difficult [51]. Profiling of ZEN metabolites may be helpful [51] because zeranol, originating from food contaminants, is expected to occur together with ZEN and its metabolites. Analysis of hair segments was also suggested, as it allows for the reconstruction of the temporal course of the exposure. In a particular case, hair analysis revealed a continuous exposure over time, which ended when zeranol was found in urine [52].

### 2.6. Note About Nomenclature

In old literature, zearalenone was mostly abbreviated ZEA and less often ZEN or ZON. The first and third acronyms do not allow deriving meaningful acronyms for zearalanone, zearala-, and zearalenols. In 1986/1988, Pompa and co-workers coined three-character acronyms encoding the structural information for zearalenone [53], zearalanone [54], and their derivatives (Figure 2). The acronyms went largely unnoticed because the papers appeared in specialized journals. In 2011, Metzler addressed the mycotoxin community at large, proposing to adopt Pompa’s acronyms in standard nomenclature [55]. Many authors and editors embraced the concept, but outdated acronyms keep popping up in experimental reports as well as in reviews.

A second issue addressed by Metzler [55] was the numbering of atoms of ZEN and its derivatives. The recommended numbering according to the IUPAC Blue Book is shown in Figure 2. Except for the heterocyclic oxygen and the methyl group, IUPAC numbering matches the traditional numbering, which is based on undecenyl lactone of β-resorcylic acid [13]. The deprecated primed locants and even apparently random numbering of atoms in the aliphatic macrocycle are still appearing in the literature; therefore, we use this opportunity to add our voice to Metzler’s recommendation.

## 3. Biosynthesis of ZEN and Its Control

The biosynthesis of ZEN proceeds via a polyketide pathway that incorporates acetate units via the acetate–malonate coenzyme A pathway [56] with polyketide synthases (PKSs) serving as key enzymes. The pathway has been primarily characterized in *F. graminearum* [57,58,59], and it involves four essential genes: *PKS13*, *PKS4*, zearalenone biosynthesis gene 1 (*ZEB1*), and *ZEB2* (Figure 3). The genes are located within a 50 kb cluster containing 11 open reading frames. ZEN biosynthesis begins with the formation of a reduced hexaketide by PKS4, which utilizes one acetyl-CoA and five malonyl-CoA molecules as substrates. PKS13 then continues the reaction by incorporating three additional malonyl-CoA molecules, resulting in the formation of a mixed reduced/unreduced nonaketide. Following two aromatic reactions, an aromatic ring and a macrolide ring with a lactone bond are generated. ZEB1, an isoamylalcohol oxidase, catalyzes the final step by oxidizing a hydroxyl group, converting β-ZEL to ZEN [59,60].

In 2015, Park and collaborators [61] demonstrated an autoregulatory mechanism of transcription in a filamentous fungus for the first time. ZEN production is autoregulated through alternative promoter usage, resulting in two *ZEB2* transcripts: *ZEB2L* and *ZEB2S*. At the N-termini of the translation product, ZEB2L contains a bZIP DNA-binding domain, whereas ZEB2S lacks this domain. ZEB2L forms homodimers, which bind to promoter regions of the biosynthetic genes, activating the entire ZEN cluster. This binding is negatively regulated by ZEB2S, whose expression depends on ZEN concentration; as ZEN accumulates, more ZEB2S transcripts are produced. ZEB2S forms heterodimers with ZEB2L, which do not bind to the promoters of structural genes of the ZEN cluster, inhibiting ZEN synthesis [61].

Cyclic AMP (cAMP)-dependent protein kinase (PKA) [62] is also involved in the control of ZEN synthesis [63]. In *F. graminearum*, two genes encode the catalytic subunits (CPK1 and CPK2) and one gene (*PKR*) encodes a regulatory subunit of protein kinase A (PKA) [62]. A detailed study of the effect of deletion of components of PKA on the levels of *ZEB2* and *ZEB2L* transcripts and ZEN production in a wild-type strain and the effect of these deletions on ZEN production in a strain overexpressing *ZEB2L* led to the hypothesis that PKA is a negative regulator of ZEN synthesis due inhibition of the transcription of *ZEB2L* but not *ZEB2S* by the active form of PKA at high cAMP concentrations [63]. The observation that the deletion of *PKR* stopped ZEN production both in the wild-type background and in a strain overexpressing *ZEB2L* rather than stimulating it (Figure 4B in [63]) indicates that the mechanism may be more complex.

The velvet complex is the second global regulator involved in the control of ZEN synthesis. The complex is formed by Ve and Lea proteins with core members VeA, VelB, and LaeA, and it plays an important role in the regulation of differentiation, response to light, and secondary metabolite production in ascomycetous fungi [64]. In *F. graminearum*, FgVelB (FGSG_01362) controls ZEN synthesis via regulation of transcript levels of *ZEB2* [65]. Deletion of the gene encoding FgLaeA, which is a component of the velvet complex in *F. graminearum*, reduced ZEN production by suppressing the expression of *ZEB2*, but overexpression of *FgLaeA* did not affect ZEN production [66].

LaeA is believed to exert its pleiotrophic effects on secondary metabolism via histone methylation [67], which is a fundamental mechanism of epigenetic control [68]. Fungi carry many histone methylases, some of which physically interact with the velvet complex [69]. The team of J. Strauss at BOKU in Vienna recently reported that FgKmt5, which is a histone methyltransferase of *F. graminearum* that methylates leucine 20 of histone 4, is necessary for the activation of ZEN biosynthesis. In contrast to LaeA, which typically affects many metabolites in the same way, the effect of *FgKmt5* appears specific for ZEN. Disruption of the gene reduced the accumulation of ZEN to 4%, while the levels of deoxynivalenol and aurofusarin remained unchanged, and fusarin C was reduced by half [70].

Attempts have been made to identify additional genes involved in the biosynthesis of ZEN in *Fusarium*. Lysoe et al. [71] identified 54 ESTs (expressed sequence tags) up-regulated under ZEN-inducing conditions, most of them related to general metabolism. None of them were located inside the ZEN cluster. The expression of *ZRA1*, encoding an ABC transporter, was stimulated by ZEN, and deletion of the gene reduced ZEN levels in culture filtrate as well as in mycelia [72]. Contradicting these results, About Ammar et al. [73] did not observe any reduction of ZEN production after disrupting *ZRA1* (designated *FGABC3* in their paper) in two strains of *F. graminearum*.

## 4. Which *Fusarium* Species Produce ZEN?

### 4.1. Disentangling the Claim That F. moniliforme or F. verticillioides Produced ZEN

The taxonomic affiliation of ZEN producers is important for considerations about the biological function(s) of the metabolite. For instance, the fungitoxicity of ZEN is only relevant in fungi that compete with or parasitize ZEN producers. Publications concerning the taxonomic affiliation of ZEN producers are marked by contradictions.

Nine *Fusarium* species have been regularly listed in the literature as ZEN producers, comprising six species generally recognized as ZEN producers and three controversial producers. The generally recognized producers (in brackets now abandoned names of perfect forms) are *F. graminearum* Schwabe (*Gibberella zeae*), *F. pseudograminearum* O’Donnell & Aoki (*Gibberella coronicola*), *F. culmorum* (Wm.G. Sm.) Sacc., *F. sporotrichioides* Sherb., *F. semitectum* Berk. & Ravenel (syn. *F. incarnatum* (Desm.) Sacc.), and *F. equiseti* (Corda) Sacc. Since the 1990s, *F. graminearum* Schwabe has been split into genealogical clades, which were gradually elevated to the level of phylogenetic species, and now constitute the *F. graminearum* species complex (FGSC) [74]. The number of species in FGSC is steadily growing, and the new species can be expected to produce ZEN. FGSC became part of the *F. sambucinum* species complex [75], which encompasses all previously recognized producers of ZEN except *F. equiseti* and *F. incarnatum*. The last two species are grouped with the *F. incarnatum-equiseti* species complex [76]. Thus, in line with a recent review [77], all *Fusarium* species known to produce ZEN belong to the *Fusarium sambucinum* species complex and the *Fusarium incarnatum-equiseti* species complex.

Some older sources also listed *F. moniliforme*, *F. oxysporum*, and *F. solani* among ZEN producers (Table 1). Note that in 1976, taxon *F. moniliforme* Snyder, Hansen & Oswald was formally replaced with *F. verticillioides* (Sacc.) Nirenberg [78], yet some strains previously assigned to *F. moniliforme* would now be referred to as *F. proliferatum* or *F. subglutinans* [79], or even *F. andiyazi*, *F. nygamai*, *F. pseudonygamai*, or *F. thapsinum* [80]. In the following, we shall focus on the claim that *F. moniliforme* produces ZEN. Table 1 lists all experimental studies known to us that supported the claim, selected studies that denied the claim, and selected reviews.

Experimental studies claiming that *F. moniliforme* or *F. verticillioides* produced ZEN can be divided into two groups. The first group is papers published till 2000. Most of these studies used one-dimensional thin-layer chromatography (TLC) for ZEN detection, which, as Smith et al. [81] showed, is prone to false positive results because *Fusarium* spp. produce metabolites that mimic ZEN on TLC plates. A second weakness is the taxonomy, because most of the studies published before 1985 have not specified how isolates were assigned to species. Doubts on the taxonomy are fueled by claims that *F. moniliforme* produced trichothecenes ([82,83]): none of the *Fusarium* species formerly known as *F. moniliforme* can produce trichothecenes, and their genomes (NCBI Acc. Nos. GCF_000149555.1, GCF_900067095.1, GCF_013396075.1) do not contain the trichothecene cluster. Some of these papers also claimed that *F. oxysporum* and/or *F. solani* produced ZEN ([84,85,86]), or their authors made such claims in other papers [87], nourishing distrust in their taxonomic assignments.

**Table 1 toxins-17-00226-t001:** Publications claiming or denying that *F. moniliforme* or *F. verticillioides* produce ZEN.

Year	ZEN Prod.	No. of Strains	Chemistry	Taxonomy ^1^	Ref.	Remark
1969	yes	2	TLC, UV, and IR spectra, GC	?	[88]	
1970	no	8	Mouse bioassay, TLC	?	[89]	
1970	no	5	Bioassay, TLC, UV spectrum	?	[90]	
1974	no	3	TLC	?	[91]	
1975	yes	1	TLC	?	[92]	
1975	yes	?	TLC, GC-ID, IR	?	[84]	
1976/77	yes/no	1/31	TLC, UV spectrum	?	[93,94]	
1976	yes/no	2/5	TLC, UV maxima	Booth 1971	[95]	
1978	yes	?	TLC, UV spectrum, bioassay	?	[83]	
1978	yes	1	TLC	?	[96]	
1981	yes	review	-	-	[97]	P. 893, 902; refers to [95]
1985	yes	?	TLC, GC-MS	?	[86]	
1985	no	4	TLC, GC-MS	Booth 1971	[98]	
1986	yes	1	HPLC-UV, MS, NMR	?	[82]	
1986	yes	review	-	-	[99]	Refers to [88] ^5^
1989	no	52 ^2^	TLC	Nelson 1983	[100]	
1990	yes	?	TLC (after agar with mycelia pressed on TLC plates)	Booth 1971, Nelson 1983	[101]	
1991	yes	1	TLC ^4^	Burgess 1983	[102]	
1991	yes/no	1/1	TLC	Booth 1971	[103]	
1991	yes/no	9/42	TLC	?	[104]	
1994	yes	1	TLC	?	[105]	
1994	yes	1	TLC	?	[106]	
1996	review	yes	-	-	[107]	Refers to [82,101,102]
1997	yes/no	7/1	TLC	?	[108]	
1997	no	28 ^2^	HPLC-UV, UV spectra	Booth 1971, Nelson 1983	[109]	
1999	yes/no	3/695	TLC	?	[110]	
2002	yes	review	-	-	[111]	No reference for the claim ^6^
2003	yes	review	-	-	[112]	Refers to [113] ^7^ and [114] ^8^
2004	yes	review	-	-	[115]	Refers to [116] ^9^
2005	yes/no	2 or 3/18 ^3^	TLC	Hoog 2000, FusKey	[117]	
2007	yes	review	-	-	[118]	No reference for the claim
2009	no	5	HPLC-MS/MS	Nelson 1983	[119]	
2010	yes	review	-	-	[120]	Refers to [121] ^9^
2012	yes	review	-	-	[122]	Refers to [117]
2013	yes	review	-	-	[123]	No reference for the claim
2013	yes	16	ELISA ^10^ (R-Biopharm)	Sequencing αTEF	[124]	
2014	yes	16	ELISA (R-Biopharm)	Sequencing αTEF	[125,126]	
2014	yes	review	-	-	[127]	P. 105, no reference
2016	yes	review	-	-	[128]	No support for the claim ^11^
2021	yes	review	-	-	[129]	Refers to [118]
2024	yes/no	2/14	ELISA (R-Biopharm)	Seq. αTEF, ITS	[130]	
2025	yes	review	-	-	[131]	Refers to [129]

^1^ Taxonomic assignment according to Booth 1971 [132], Nelson 1983 [133], Burgess 1983 [134], de Hoog 2000 [135], FusKey [136]; ^2^ sum of strains designated *F. moniliforme*, *F. proliferatum*, and *F. subglutinans*; ^3^ cf. p. 278 and Table 1; ^4^ GC-MS was used in the study but ZEN was detected only by TLC; ^5^ further papers cited do not support the claim; ^6^ claim that *F. moniliforme* produces 1–19 ppm ZEN given without source; ^7^ textbook of medical mycology; ^8^ review in a veterinary journal; ^9^ referenced papers do not support the claim; ^10^ Enzyme-Linked Immunosorbent Assay; ^11^ none of the twelve references attached to Table 1 support the claim, and some (e.g., [137]) do not list *F. moniliforme*/*F. verticillioides* among ZEN producers.

The second group consists of four experimental studies supporting the claim that were published after 2000. The taxonomic assignments in the first of these reports were based on an atlas of clinical fungi [135], which is unsuitable for fungal isolates from maize. The remaining three studies determined partial sequences of the translation elongation factor 1α gene; therefore, their taxonomy was likely correct. To paraphrase W. Marasas, the problem was the chemistry. ZEN was detected by ELISA, which was not an adequate method, as we discuss in the following section.

Our literature review focuses on *F. moniliforme/F. verticillioides*, but the situation is similar for *F. oxysporum* and *F. solani*. The only report by leading taxonomists that assigned ZEN to one of these species appeared in the compendium “Toxigenic *Fusarium* Species: Identity and Mycotoxicology” by Marasas, Nelson, and Toussoun [31], where two *F. oxysporum* strains from Japan were reported to produce ZEN. In 2018, the work was re-evaluated by current taxonomists in a vast experimental effort, which was possible because most strains were still available [138]. The results showed that neither of the two *F. oxysporum* strains (one of which was re-classified as *F. commune*) produced ZEN.

### 4.2. Misleading Results of ELISA Without Proper Negative Controls

In research on ZEN, incorrect use of ELISA led to wrong conclusions in two areas: production of ZEN by *F. moniliforme/F. verticillioides*, and ZEN as an endogenous metabolite of plants. We believe that in both cases, false positive results of ELISA lead to wrong conclusions.

Fungal and plant extracts are complex matrices that may contain components that nonspecifically bind to antibodies. Even if some strains of the same fungal species or accessions of the same plant species test negative, they cannot be used as negative controls because metabolic profiles among strains and accessions vary. A positive result for strain A, obtained along with a negative result for strain B of the same species as a negative control, does not prove that strain A produced ZEN. Matrix components that are nonspecifically binding to antibodies may have been present in strain A but are missing from strain B. These considerations call the results of the experimental studies since 2013 listed in Table 1 and the publications claiming that over 30 plant species produced ZEN (Section 6) into question.

ELISA for ZEN also appeared in a 2006 study of genes involved in ZEN synthesis [57]. *F. graminearum* strains in which these genes were disrupted should not produce any ZEN. ZEN was quantified by ELISA, with uninoculated rice utilized as a negative control. The authors reported that “the disrupted transformants produced 0.9 ± 0.2 µg/g ZEA, a much lower level and less than that found in the uninoculated rice control extracts (1.7 ± 0.2 µg/g)”, but TLC used in addition to ELISA showed that the transformants have not produced any ZEN. The positive ELISA results may have originated from contamination of rice with ZEN, which is common [139,140], or from unknown matrix components interfering with the assay. The decrease of the signal in fungal cultures as compared to uninoculated rice could be due to adsorption or metabolism of ZEN or the interfering components. The authors rightly ignored the ELISA results, relying on TLC and concluding that the disrupted transformants have not produced ZEN.

Today, the gold standard for ZEN analysis is HPLC-MS/MS (HPLC coupled with tandem mass spectrometry). Less common yet equally reliable is HPLC-HRMS (HPLC coupled with high-resolution mass spectrometry). GC-MS is coming out of fashion, but the combination of high chromatographic resolution and mass specificity also allows for highly reliable identification of ZEN. TLC cannot compete with these methods, but after an adequate cleanup, multiple developed high-performance TLC [141] and two-dimensional TLC [84,142] allow specific detection of ZEN with limits at the low nanogram range. These methods should be used instead of ELISA for unconventional matrices.

### 4.3. Perpetuation of Disputed Claims in Reviews

Since 2000, nine reviews and a Wikipedia article perpetuated the long-disproved claim that *F. moniliforme* or *F. verticillioides* produced ZEN. In a Wikipedia article [131], the claim survived an update in October 2024. It is instructive to trace its origin. Wikipedia [131] refers to a single review [129], which refers to a monograph of veterinary toxicology [118], which made the claim without any substantiation.

Leading experts have consistently disputed the claim that *F. moniliforme/F. verticillioides* produces ZEN, beginning with the compendium by Marasas, Nelson, and Toussoun in 1984 [31]. In 1989, Ulf Thrane criticized reports about ZEN production by *F. moniliforme/F. verticillioides*, *F. oxysporum*, or *F. solani* as insufficiently documented [143]. In 2006, Ann Desjardins wrote in her excellent monograph, “…The author’s conclusion is that the most convincing and current evidence argues that zearalenones are not produced by *F. verticillioides* and related species” [144]. In 2017, Gary Munkvold consented, “Fungi in the *F. oxysporum*, *F. solani*, and *F. fujikuroi* species complexes are not considered to be zearalenone producers” [145] (*F. fujikuroi* sp. complex holds all species formerly referred to as *F. moniliforme*). That disputed claims are promoted in respected journals, despite contrary evidence and overwhelming expert opinion, highlights a quality control problem in scientific publishing.

### 4.4. Search for Homologues of PKS4 and PKS13 in Fusarium Genomes

The lack of a particular metabolite can be proved by demonstrating the absence of the gene required for its synthesis from the genome. The genomes of several *Fusarium* spp. have been sequenced, including the disputed ZEN producers *F. verticillioides*, *F. oxysporum*, and *F. solani*. Search for homologues of PKS4 and PKS13 from *F. graminearum* in the genomes of the three disputed species failed to find any such gene (Table 2).

Genome data from two recently established species of the *Fusarium incarnatum-equiseti* species complex, namely *F. flagelliforme* and *F. clavum* [76], became available recently, and they are included in Table 2. Nothing is known about the production of ZEN by these species. We have identified genes whose products are putatively involved in ZEN synthesis in both species. The GeneBank acc. nos. of the translation products are CEG04361 and CEG04510 for *F. clavum* and RFN53982 and RFN53983 for *F. flagelliforme*. In *F. clavum*, the translation products are annotated as unknown proteins. In *F. flagelliforme*, they are annotated as polyketide synthases. Their homology with polyketide synthases involved in ZEN synthesis in *F. graminearum* (above 90% identity of amino acid residues for PKS4 and above 87% for PKS13) strongly indicates that *F. clavum* and *F. flagelliforme* are ZEN producers.

### 4.5. Can We Be Certain That No Strain of the Disputed Species Produces ZEN?

The lack of convincing experimental support does not eliminate the possibility that strains of the disputed species producing ZEN exist in nature. Fungi are able to acquire secondary metabolite pathways by horizontal gene transfer [146]. Comparative genome analysis in the *Fusarium incarnatum-equiseti* species complex revealed gene clusters acquired by horizontal transfer that became part of standard genetic equipment [147]. Despite this possibility, claims that *F. verticillioides*, *F. oxysporum*, or *F. solani* can produce ZEN remain unfounded until such strains are found and their capability to produce ZEN is conclusively demonstrated.

### 4.6. Unexpected Discovery of Genes for ZEN Biosynthesis in the F. verticillioides Genome

Recently, both PKS genes involved in ZEN synthesis have been reported in the genome of a particular strain of *F. verticillioides* [148]. The gene allegedly encoding the nonreducing PKS involved in the synthesis of ZEN was named ZEA1, and the gene encoding the reducing PKS was named ZEA2. These are the names used by one of the teams that identified these genes in *F. graminearum* [57]; the genes were later renamed PKS13 and PKS4 [58,59].

Comparison of protein sequences shows that ZEA1 of [148] is in fact *bik*1 (Acc. No. S0DZM7), which is a key gene of bikaverin synthesis [149], and ZEA2 of [148] is *FUB1* (Acc. No. W7MT31), which is involved in fusaric acid synthesis [150].

The products of *bik1* and *FUB1* share no homology with the PKSs involved in ZEN synthesis (Appendix A). Why have they been assigned to ZEN synthesis? We speculate that a historical coincidence was involved because the PKS responsible for the first step of bikaverin synthesis was also named PKS4 [149]. The name has been used for some time (e.g., [151]), but the gene was later renamed *bik*1 [152,153]. To add to the confusion, the accession number of the gene in [149] was given as AF278141.Bl (p. 140), which now designates a hypervariable region in human mitochondrial DNA. A BLAST (Basic Local Alignment Search Tool) search against UniProt or consultation of an authoritative review would have alerted the authors that something was wrong.

### 4.7. Remembering Wally Marasas

We finish this section by quoting W. Marasas, a pioneer of research on *Fusarium* mycotoxins (Walter Friederich Otto Marasas, 1941–2012). In his entire career, Marasas promoted good scientific practices, and he was not shy about criticizing low standards in his field. He was particularly concerned about questionable mycotoxin-species assignments, and his colleagues and participants at *Fusarium* Laboratory Workshops became acquainted with his sarcastic comment (Figure 4). One of the reasons why he, Nelson, and Toussoun wrote their legendary monograph *Toxigenic Fusarium Species* [31] was to rectify wrong mycotoxin-species assignments. In the Introduction, the authors wrote: “This situation has led to great confusion in the extensive literature on *Fusarium* mycotoxicology…, several *Fusarium* toxins have been named for mis-identified producing species…, chemical and pathological studies have been attributed to incorrectly named species”.

Applying Marasas’s statement to the claim that *F. moniliforme*/*F. verticillioides* produced ZEN, Table 2 indicates that the taxonomy was probably wrong in many old studies, while the chemistry was wrong in the last three studies. With respect to the recent claim that *F. verticillioides* harbors genes involved in ZEN synthesis [148], Marasas’s statement can be extended to encompass wrong bioinformatics.

## 5. HYPOTHESIS 1: ZEN Is a Fungal Hormone Controlling Sexual Development

### 5.1. Foundation and Experimental Support

The hypothesis that ZEN acts as a hormone in *Fusarium* spp. was raised in 1968 by Cesaria Eugenio while working on her Ph.D. thesis at the University of Minnesota. Eugenio observed that the application of small amounts of ZEN to cultures of *F. graminearum* on agar induced mass production of perithecia [156]. Three years later, M. Wolf, who was working at the same laboratory, confirmed her results in his thesis [157]. In the literature, however, the hypothesis that ZEN is a sexual hormone of *Fusarium* spp. is attributed to R.R. Nelson from Pennsylvania State University, based on his presentation at a conference in Hawaii in 1971 and a 20-page-long report in the conference proceedings [29]. As an extension of his preliminary work [158], Nelson studied the effect of ZEN on 45 species of sexually reproducing fungi and found stimulation by low concentrations of ZEN in 40 species. High concentrations of ZEN inhibited sexual reproduction as well as growth. Nelson never investigated the effect of ZEN on *Fusarium* spp., which would be necessary to uphold the hormone hypothesis. In the discussion that followed Nelson’s presentation, Mirocha revealed similar results on *F. graminearum* that had not yet been published at the time. The discussion was recorded and published along with Nelson’s report [29].

Nelson’s hypothesis fell on receptive ears. With the discovery of steroid hormones in Achlya, the concept of fungal hormones was becoming popular [159]. Researchers at Pardue University were the first to support the hypothesis by reporting “close association” between ZEN production and perithecia formation in *F. roseum* “Graminearum” [89], but major support came from the University of Minnesota. The key finding from Mirocha’s lab was that dichlorvos (organophosphate insecticide) inhibited both ZEN accumulation and perithecia formation, and that simultaneous application of ZEN prevented the inhibition of perithecia [160]. Encouraged by these results, Mirocha’s lab launched a series of studies on the phenomenon. Firstly, they confirmed Eugenio’s finding that perithecia formation was stimulated by small amounts but inhibited by large amounts of ZEN [161]. In the same study, they found that a hydroxy- or keto-group on C7 (in their paper C’6) was necessary for the stimulation, and the double bond C11-C12 was necessary for the inhibition of perithecia formation. Cyclic 3′,5′-adenosine monophosphate (cAMP) added to the culture stimulated both perithecia formation and ZEN synthesis in *F. graminearum*, prompting speculations that the effect of cAMP on sexual reproduction was mediated by ZEN [162]. Mirocha’s lab also showed that ZEN binds to proteins extracted from *F. graminearum*, and they purified a putative ZEN receptor [163] and showed that it also binds mammalian steroid hormone 17-β-estradiol [164]. Comparison of the effect of the *cis*-isomer and *trans*-isomer of ZEN on *F. graminearum* and rats revealed another similarity: the *cis*-isomer of ZEN exerted a stronger estrogenic effect than the *trans*-isomer in mammals, and only this isomer stimulated perithecia production in *F. graminearum.* The inhibitory effect exerted by the *trans*-isomer at high concentrations was absent [164].

### 5.2. Criticism

In the 1980s, Mirocha’s laboratory regarded the hypothesis as confirmed. They wrote “Wolf and Mirocha [160,161] unequivocally demonstrated the role of trans-zearalenone in perithecia development” [164] and “zearalenone is a sex hormone produced by various species of *Fusarium*” [56]. The community received the hypothesis positively (e.g., [165,166]). The fact that *F. culmorum* produced large quantities of ZEN (according to some reports even more than *F. graminearum*, e.g., [94,167,168]) without having a sexual stage was not taken into account.

The first doubts arose during comparative analysis of *F. graminearum* strains and populations. The production of ZEN by *F. graminearum* strains was expected to correlate with the ability to form perithecia, which was not observed. The lack of perithecia in strains accumulating high levels of ZEN was explained by the inhibitory effects exerted by ZEN at high concentrations [165], but the formation of perithecia in the absence of ZEN [169] could not be explained. These results came from the investigation of *F. graminearum* populations labeled Group 1 and Group 2. Group 1, which was later elevated to a species level as *F. pseudograminearum* [170], did not produce perithecia, while Group 2 produced abundant perithecia on suitable media such as carrot agar. Mirocha predicted that only Group 2 would produce ZEN, which was shown to be wrong [169]. An investigation of 93 isolates showed that both Group 1 and Group 2 produced ZEN at different levels [30]. Contrary to the hypothesis, isolates not producing ZEN as well as isolates producing inhibitory levels formed perithecia. The paper, co-authored by Mirocha, factually falsified the hypothesis that ZEN is a sexual hormone in *F. graminearum* [30].

In the following years, the hypothesis was often presented together with contradicting findings, and it was referred to as undecided [171,172].

A final blow to the claim was delivered when the biosynthesis pathway for ZEN in *F. graminearum* was disrupted by genetic engineering. Three teams achieved this feat in 2005 and 2006. Kim et al. at Seoul National University identified polyketide synthase genes PKS4 and PKS13 that are essential for ZEN synthesis, and they deleted both genes by replacing them with a hygromycin resistance cassette [59]. The resulting strains formed perithecia normally. Lysøe et al. from collaborating Danish and Norwegian labs disrupted PKS4 by an insertion and, too, have not recorded any effect of the loss of ZEN production on perithecia formation [58]. Finally, Gaffoor et al. at Michigan State University and the U.S. Department of Agriculture in Peoria disrupted both PKS genes as part of a larger effort targeting 15 PKS genes in *F. graminearum* [173]. The disruption of both PKS13 and PKS4 genes did not affect perithecia formation, and the perithecia discharged ascospores normally.

The disproved hypothesis continues to pop up occasionally. A review about fungal sensing published in 2016 included fungal mating among potential functions of ZEN [174], and in 2019, researchers studying the effect of ZEN on anther cultures of wheat wrote that ZEN “is an endogenous regulator of the sexual stage of development of fungi” [175]. The source they cited is a book chapter [176], which stated “this component is believed to act as an endogenous regulator of the sexual stage of development of their producer fungi” without providing any reference.

### 5.3. Can the Results Be Explained Without Invoking a Hormone Hypothesis?

The hypothesis was disputed, but Nelson’s results and reports from Mirocha’s lab remain: Why was the sexual reproduction in many fungal species stimulated by ZEN? We offer an explanation that is based on the fungitoxicity of ZEN and the hypothesis about the partition of metabolic resources between growth and reproduction. Harmful treatment, such as wounding [177,178], oxidative damage [179], and injury caused by fungicides [180], growth-inhibiting agents [181], or detergents [182] stimulates fungal sporulation. These observations can be explained by the hypothesis about optimal allocation of resources, which has been developed primarily for plants [183]. ZEN is strongly fungitoxic ( [184]; see also Section 8). In line with the theory of optimal allocation of resources, we suggest that fungi exposed to ZEN sense harmful conditions; therefore, they allocate more resources to reproduction. In plants, this strategy has been dubbed “big-bang reproduction” [185]. We propose that the same reasoning applies to fungi.

## 6. HYPOTHESIS 2: ZEN Is a Plant Hormone

### 6.1. Stimulatory Effects of ZEN on Plants

*Fusarium* spp. produce phytohormones auxin, cytokinin, gibberellins [186,187], and ethylene [188]. Hormone-like effects of ZEN on plants were for the first time reported by Mirocha’s lab in 1968. They published their results on tobacco callus together with the results on animals in the journal *Cancer Research* [189], which could be why the work was overlooked. The next mention appeared in the proceedings of the conference Plant–Parasite Interactions in Hawaii in 1971 [29], which are well known because of Nelson’s report about the stimulation of sexual reproduction in fungi by ZEN (Section 5). The passage about the effect of ZEN on plants appeared only in the discussion, which was recorded and published together with the proceedings. There was a comment on Nelson’s observation of cytokinin-like activity of ZEN in tobacco callus, which Nelson excluded from his written report. Furthermore, Mirocha testified that his co-worker Linsmaier-Bednar observed the stimulation of the growth of tobacco callus and shoots by ZEN. It was not until 1978 that the next report on the topic appeared, which was a short article in a journal of the International Seed Testing Association. The authors described moderate stimulation of the growth of maize embryos by ZEN [190]. Afterward, the topic was abandoned for 20 years.

Since the 1990s, Fan-Jing Meng at Beijing Agricultural University investigated stimulatory effects of ZEN on plants, joined a few years later by researchers at the Polish Academy of Sciences in Krakow and at universities in Krakow, Warsaw, and Poznan. Only one study came from another laboratory. In 1996, McLean at the University of Natal, South Africa, reported stimulation of the growth of maize embryo and primary roots and shoots [191] by ZEN. These results are summarized in Table 3. We also cite an astounding recent claim that ZEN lactonase secreted by *C. rosea* in soybean roots may help “stabilize plant hormone levels” [192].

**Table 3 toxins-17-00226-t003:** Stimulatory effects of low concentrations of ZEN ^1^ on plant growth and development.

Year	Plant	Effect	Ref.
1968	Tobacco	Growth of callus, formation of shoots and roots	[189]
1978	Maize	Growth of embryo	[190]
1991	*Lemna perpusilla*	Enhancement of flowering	[193]
1993	*Lemna gibba*	Enhancement of flowering	[194]
1996	Maize	Growth of embryo, primary root, and shoot ^2^	[191]
1998	Wheat	Generative growth	[195]
1998	Wheat	Growth of haploid embryos	[196]
1999	Wheat	Embryogenesis of wheat callus	[197]
2003	Wheat	Growth of haploid embryos after pollination with maize	[198]
2006	Wheat	Wheat production	[199]
2006	Soybean	Soybean production	[200]
2009	Soybean, wheat	Photosynthesis rate, seed number, and weight	[201]
2010	Soybean, wheat	Regeneration of plants from callus	[202]
2010	Winter wheat	Acceleration of vernalization	[203]
2011	Soybean and wheat	Efficiency of photosynthesis ^3^	[204]
2017	Maize, wheat, sorghum	Tolerance to osmotic stress	[205]
2019	Wheat	Microspore embryogenesis	[175]
2021	Legumes	Yield, protein content, sugar content	[206]
2022	*Tetrastigma hemsleyanum*	Growth of shoots and roots	[207]
2023	*Tetrastigma hemsleyanum*	Root growth and miRNA accumulation	[208]
2024	Soybean	ZEN lactonase may stabilize the plant hormone levels	[192]

^1^ High concentrations of ZEN have sometimes also been tested, and in most cases, they inhibited growth and development; ^2^ in contrast to other studies, McLean observed stimulation of growth at ZEN concentrations labeled as high (10 and 25 µg/mL), while a low concentration (5 µg/mL) inhibited growth; ^3^ in wheat, ZEN increased the efficiency of photosynthesis only under strong illumination, while in soybean, the effect was observed at both light intensities.

It has been known since the 1930s that steroidal estrogens from animals stimulate plant growth [209] (for those who searched [209] for estrogens in vain, it was estrone, which the authors called folliculin. Butenandt’s term folliculin has been abandoned, except by Russian authors, and later re-assigned to a protein associated with kidney cancer). Stimulatory effects of steroids from higher vertebrates on many plants have been reported (reviewed in [210]). Because ZEN resembles steroidal estrogens by binding to estrogen receptors in mammals [211], it is perhaps not surprising that ZEN exerts similar stimulatory effects on plants as steroidal estrogens.

Inspired by the hormone-like effect of ZEN on plants, a surprising hypothesis was proposed and later supported by data, becoming a claim.

### 6.2. ZEN as an Endogenous Plant Regulator

The hypothesis was raised in 1986 by Fan-Jing Meng and her team at Beijing Agricultural University, and it was explored at least until 2000, with initial publications in Chinese journals. Since 1995, they have published in international journals including *Journal of Plant Physiology*, *Flowering Newsletter* (now *Journal of Experimental Botany*), and *Plant Growth Regulation*. In their last paper on the topic, published in 2000, they monitored endogenous ZEN in tobacco plants during flowering, and they suggested that ZEN was involved in flower bud development [212].

Since 2001, a Polish team led by Biesaga-Kościelniak at the Institute of Plant Physiology of the Polish Academy of Sciences in Krakow, who have been studying stimulatory effects of ZEN on plant growth and development (see previous section), embraced the claim that ZEN was an endogenous plant hormone. Table 4 shows relevant publications from both laboratories on this topic.

In reviews from both labs, endogenous ZEN has been claimed to occur in over 30 plant species [176,222]. Apart from the investigation of endogenous ZEN, both labs studied the effects of the treatment of plants with ZEN, and they consistently labeled ZEN as a plant hormone in these reports, listing ZEN along with indole-3-acetic acid and gibberellic acid (e.g., [205,227]).

### 6.3. Criticism

If ZEN were an endogenous plant metabolite, it should have been found in plant extracts in at least some of the numerous metabolomic studies on plants. Wheat, tobacco, maize, and oilseed rape, which belong to the species claimed to accumulate endogenous ZEN, have been subjected to extensive untargeted metabolomic analyses. To our knowledge, none of these studies found endogenous ZEN.

The authors of a study on the uptake and transformation of ZEN by wheat plants, being aware of the hypothesis, tested untreated plants for the presence of ZEN. Neither at the beginning nor at the end of the experiment did they find ZEN. They wrote, “At the tested conditions (LOD 0.35 ppb) we may exclude the presence of endogenous ZEN in wheat plantlets” [228].

How was ZEN detected in the studies that claimed that ZEN was an endogenous plant metabolite? In the first report from Meng’s lab, a ZEN-like substance was detected by TLC, and its UV spectrum was shown to resemble the spectrum of ZEN [213]. Later, the authors confirmed the identity of the metabolite with ZEN by comparing its retention time in HPLC with an authentic standard and by determining its molecular weight and daughter spectrum by mass spectrometry [215,216]. The data presented in these papers do support the claim that the sample contained ZEN, but we believe that the source of ZEN was likely a ZEN-producing *Fusarium* sp. that spontaneously infected the plants because the samples were obtained from seedlings of winter wheat growing in the field. In the subsequent publications, Meng’s group mainly used ELISA, which was developed in their lab with rabbit antibodies [229], and sometimes TLC [212,222]. We believe that their interpretation of the ELISA results fell victim to the same fallacy as ELISA support for the production of ZEN by *F. moniliforme*/*F. verticillioides* (Section 4). Mass spectrometry was used in the analysis of maize and wheat grains [223], but contamination of the samples with *Fusarium* spp. was neither checked nor excluded.

Based on the absence of ZEN from plant metabolomes recorded in numerous metabolic studies in the last decade, the low specificity of the methods used in many studies that reported endogenous ZEN in plants (TLC and ELISA), and the lack of a check for contamination in the few studies that used advanced analytical methods, we believe that the claim that plants produce ZEN is unsubstantiated.

Publications that portray the disputed hypothesis as an established fact [206] or quote it uncritically [208] still pop up in spite of a growing number of reported plant metabolomes, none of which revealed the presence of ZEN.

## 7. HYPOTHESIS 3: ZEN Is a Virulence Factor of *Fusarium* spp.

### 7.1. Origin and Support

To our knowledge, the hypothesis that ZEN is a virulence factor in plant infection was mentioned in two publications [230,231], but only the latter endorsed it. A third study [232] reported results that indicated a possible role of metabolites similar to ZEN in plant infection.

The first study [230], carried out in the lab of F. Trail at Michigan State University, focused on the resistance of inbred lines of maize to stalk infection with *F. graminearum*. The authors used a wild-type strain of *F. graminearum* and its mutants ∆tri5 and ∆zeal∆zea2, in which deoxynivalenol (DON) or ZEN synthesis, respectively, was disrupted. Twenty inbred lines were inoculated by placing agar overgrown with a mycelium over small punctures in the stalks, necrotic lesions were monitored, and the area under the disease progress curve was calculated. The ZEN-nonproducing mutant caused smaller lesions than the wild-type strain in some inbreds but not in others. The authors wrote cautiously, “Four and six inbreds that were susceptible to PH1 were moderately resistant to ∆tri5 and ∆zeal∆zea2, respectively, suggesting that DON and ZEA could be virulence factors in maize”.

The second study [231] was carried out by an Austrian team under the leadership of G. Adam from the University of Natural Resources and Life Sciences (BOKU) in Vienna. The authors investigated the effect of ZEN on heat shock protein 90 (HSP90) in *Arabidopsis thaliana* and the glycosylation of ZEN by the plant. Glycosylated ZEN, ZELs, and radicicol did not inhibit HSP90. The authors concluded that “ZEN had a very prominent target in plants”, and that inactivation by glycosylation “may explain why there is little evidence for a virulence function of ZEN in host plants.” They also cited selected results of [230] to support the hypothesis, see section *Criticism* below for details. The hypothesis was presented cautiously (“This study suggests that ZEN may have a role in virulence…”), and its weaknesses were addressed (“…it is surprising that loss of ZEN production in *Fusarium* does not lead to clearly reduced virulence”). The authors also acknowledged an alternative hypothesis that ZEN is a defense metabolite and even cited supporting evidence. In the closing passage, however, they appeared to present the virulence hypothesis as established and the defense hypothesis as a mere option: “*Apart from a role as a virulence factor* in plant disease development, ZEN *may also have an ecological role* in preventing competing fungi from colonizing substrates” (emphasis added).

An interesting result of [231] was that β-ZEL inhibited HSP90 more strongly than ZEN, while α-ZEL was almost inactive. This is opposite to the inhibition of fungal growth by ZELs [184] and to their estrogenic effects [233]. The authors speculate that increasing the conversion of ZEN to α-ZEL could be used in resistance breeding, provided ZEN is a virulence factor. They cite results showing that *A. thaliana* and wheat plants rapidly converted ZEN to β-ZEL and, to a smaller extent, α-ZEL, and speculate that shifting the conversion towards α-ZEL might increase plant resistance to infection. Notwithstanding our critique of the virulence factor hypothesis, we found it interesting to compare the described in vitro results with the ratio of α- and β-ZEL in grain in the field. Published data indicate that wheat grains typically accumulate more β-ZEL, while maize grains accumulate more α-ZEL (Appendix B and Table A1).

A third study to mention, conducted under the direction of M.W. Sumarah at Agriculture and Agri-Food Canada in London, Canada, investigated metabolites of pathogenic and apathogenic species of Ilyonectria [232]. Metabolites produced solely by virulent strains included radicicol and compounds putatively identified as pochonins and monocillins, all of which, like zearalenone, are RALs (resorcylic acid lactones). The authors suggested that these RALs should be tested in future experiments as pathogenicity factors. Remarkably, virulent strains produced more metabolites than avirulent strains, and metabolites produced by both types of strains were produced in larger quantities by virulent strains. Furthermore, chemical profiles of virulent species *I. mors-panacis* and *I. robusta* were indistinguishable, which the authors interpreted as indicating that “perhaps secondary metabolites were important to the virulence of these species”. We offer an alternative explanation (see below). The authors also considered a defense function of RALs, suggesting that RALs may assist Ilyonectria in colonizing the roots by suppressing other pathogens in the soil.

### 7.2. Criticism

#### 7.2.1. Inhibition of Plant Heat Shock Protein by ZEN and Detoxification of ZEN by Plants [231]

The publication [231] has been cited four times, but only one of the citing papers mentioned the hypothesis about ZEN as a virulence factor [234]. The authors, with G. Adam among them, acknowledged the established defense function of ZEN, while calling the virulence function of ZEN uncertain.

The key argument of [231] is the inhibitory effect of ZEN on plant HSP90. This was not surprising because many RALs [235,236,237], including ZEN [238,239], inhibit mammalian HSP90, and ZEN also binds to human HSP90 [240]. HSP90 of *A. thaliana* studied in [231] belongs to class A, which is highly conserved across kingdoms [241]. We argue that the inhibition of plant HSP90 by ZEN is as incidental as is the inhibition of mammalian HSP90 or mammalian mitogen-activated protein kinases [242,243]. They do not reveal anything about the biological function of ZEN for the same reason that stimulation of plant growth (see previous section), estrogenic effects on mammals (Section 2), and stimulation of fungal sexual reproduction (Section 5) do not. As ecologists have always emphasized, proof of a biological function of any metabolite strictly requires demonstrating that the metabolite increases the producer’s fitness [244,245].

The second argument is the detoxification of ZEN by plants, which could be seen as a defense response. We argue that the coincidence of inhibition and detoxification does not imply function. Similarly, the coincidence of harmful effects of ZEN on animals and detoxification of ZEN by animals [246,247] does not imply that the effect of ZEN on animals is functional. Most enzymatic detoxification activities towards mycotoxins discovered so far are likely accidental, but even a functional detoxification that protects its host from toxic effects and underlies positive selection does not imply that the effect of ZEN on the host is functional, enhancing the fitness of ZEN producers.

The authors noted that benzenediol lactones, to which ZEN belongs, are widespread among fungi, and they speculate that several of these metabolites could potentially play a role in plant–pathogen interactions. With reservation (Section 7.4), we argue that the widespread occurrence of RALs among fungi rather indicates that their function is ancient and related to a vital survival mechanism shared by all fungi, not a few species that colonize living plants.

The last argument that we address concerns the production of ZEN in planta. Several labs have shown that the inactivation of ZEN synthesis in *F. graminearum* does not affect the infection of wheat and barley. To reconcile these results with their hypothesis, the authors posited that ZEN biosynthesis genes were not expressed under these conditions, inferring that the conclusion that ZEN plays no role in virulence might be premature. We draw an opposite conclusion: if ZEN is not produced during infection, it cannot play any role in virulence.

#### 7.2.2. Infection of Maize Stalks with ZEN-Nonproducing Mutant [230]

The results of the study of the infection of stalks of 20 maize inbred lines with a *F. graminearum* strain and its DON-nonproducing and ZEN-nonproducing mutants [230] generated diverse and partly contradictory results. The authors’ cautious inference that the results “suggest that DON and ZEA could be virulence factors in maize” was based on the observation that DON mutants caused weaker infection in 4 out of 20 inbreds and ZEN mutants caused weaker infection in 6 out of 20 inbreds. The response of inbred lines to virulence factors may differ, but this fact cannot explain why both mutants caused *larger* lesions than the wild-type strain in some inbreds. The only plausible explanation we can think of is that part of the differences among inbreds reflected statistical variation rather than genuine differences among genotypes. The authors of [231] pointed out that disease symptoms caused by the ZEN mutants in [230] were reduced by about 40% (in fact, it was just 30%) and that in one inbred line, the virulence of the ZEN mutant was reduced more than the virulence of the DON mutant. We counter that the large variation among inbreds facilitated cherry-picking any outcome, including strains with disrupted ZEN or DON synthesis, appearing more virulent than the wild-type strain.

#### 7.2.3. Production of RALs by Virulent Strains of Ilyonectria spp. [232]

Finally, we look at the study that suggests that certain RALs similar to ZEN may act as virulence factors in Ilyonectria spp. [232]. The hypothesis was based on the observation that these RALs were produced only by virulent strains, but it is important to note that (i) the entire metabolomes of virulent and avirulent strains were separated, (ii) virulent strains produced larger number of metabolites than avirulent strains, and (iii) virulent strains produced larger quantities of metabolites common to both virulent and avirulent strains. These findings are remarkable because virulent and avirulent strains of plant pathogens are typically very similar and often phenotypically indistinguishable, differing only in a few virulence and/or avirulence factors.

The crucial point is that the metabolic profiles labeled as belonging to virulent and avirulent strains were obtained from different fungal species. This is not evident from the description of the analysis because the samples are always referred to as virulent and avirulent strains. In the description of the PCA analysis, the members of the virulent and avirulent strain groups are listed by their ID numbers. One has to scroll to the end of the paper to find out that all the virulent strains belonged to the two species *I. robusta* and *I. mors-panacis*, while all the avirulent strains belonged to other species: *I. torresensis*, *I. rufa*, *I. estremocensis*, and *Neonectria obtusisporum*. Thus, the metabolomic clusters reflected differences between the two groups of species rather than virulent and avirulent strains.

The second issue is that the method used for metabolite extraction was a micro-scale screening procedure [248], which lacked normalization of the extracts to fungal biomass. Without normalization, quantitative comparison of metabolite production is not possible. It is plausible that the two species that apparently produced larger amounts of common metabolites (and that also happened to be virulent) grew faster, accumulating more biomass. This also explains why a larger number of metabolites were detected in these extracts, because lower concentrations obtained from smaller amounts of biomass are more likely to fall below detection limits.

The authors hypothesize that RALs produced by pathogenic species of Illionectria may indirectly assist their producers in colonizing ginseng roots by inhibiting other pathogens in soil, improving their chance of surviving in the soil, and eventually infecting ginseng. We think that this is likely, considering the extensively documented antifungal activities of RALs (Section 9.3). The authors also speculate that RALs of Illionectria may act as virulence factors, and they suggest that enzymes that degrade ZEN may also inactivate other RALs, and may be used to test the hypothesis. We believe that the chance that ZEN lactonases degrade related RALs is high, and that the concept deserves testing. Until then, the function of RALs as virulence factors in this pathosystem remains speculative.

### 7.3. How to Recognize That a Fungal Metabolite Is a Virulence Factor?

Before gene disruption techniques became available in fungi, the assessment of the role of a fungal metabolite in pathogenicity (ability to cause disease), virulence (the degree of disease symptoms), and aggressiveness (quantitative measure of colonization) has relied on indirect evidence and correlations (Table 5).

These approaches are still useful today because they help select candidate genes for disruption experiments and provide support for hypothesis testing. Table 5 summarizes indirect as well as direct evidence for and against the role of ZEN production by *F. graminearum* in the infection of wheat ears. For a comparison, the results for DON are also shown, which is an established virulence factor of *F. graminearum* in wheat [274].

### 7.4. Disclaimer

Experimental proof of a fungal metabolite’s role in virulence, or its absence, holds only for the pathosystem in which the proof was obtained. We argued that ZEN does not contribute to the virulence of *F. graminearum* in the ears of wheat and barley, and we will argue that the fundamental biological function of ZEN and related fungal macrolactones is to antagonize fungal competitors and mycoparasites. This does not rule out the possibility that in a different pathosystem, perhaps one that has yet to be discovered, ZEN may contribute to virulence, apart from its primary function as an agent of chemical defense. We are not aware of any fungal specialized metabolites with more than one rigorously proven biological function, but there is no reason to expect that evolutionary tinkering, which has been demonstrated for enzymes and signal pathways, would halt at specialized metabolites.

## 8. HYPOTHESIS 4: ZEN Is a Defense Metabolite Protecting *Fusarium* from Mycoparasites and Suppressing Competitors

### 8.1. Supporting Observations

Three observations can be seen as support for the hypothesis: (i) inhibition of fungi by ZEN, (ii) induction of ZEN synthesis in mixed cultures of *F. graminearum* or *F. culmorum* with competing fungi, and (iii) detoxification of ZEN by fungi that antagonize ZEN producers. Individually, these observations hold little weight, but when considered collectively, they strongly suggest that the hypothesis warrants testing.

#### 8.1.1. Inhibition of Fungal Growth by ZEN

The inhibitory effects of ZEN on fungi have been observed by Nelson et al. at a conference [158], three years after the discovery of ZEN. In his well-known paper that laid the foundation for the sexual hormone hypothesis, Nelson mentioned the inhibition of fungal growth by ZEN in passing, and in the discussion after his report, he stated that the effect was not specifically examined [29]. These results were superseded by more interesting biological activities, namely estrogenic and anabolic effects (Section 2.1) and stimulation of fungal sexual development (Section 5). Later, the antifungal effects of ZEN were observed in studies that used ZEN as a control in bioprospecting natural products, for instance [275]. Despite receiving little attention, these results further substantiated the inhibitory effect of ZEN on fungi. The inhibition of the growth of 11 filamentous fungi and fungus-like stramenopiles by ZEN at 10 µg/mL was reported in the work that established the function of ZEN lactonase in *Clonostachys rosea* (previously *Gliocladium roseum*) [184], discussed below. The growth of six Trichoderma species on agar was significantly inhibited by ZEN at 4 µg/mL [276]. Ten micrograms ZEN per milliliter completely blocked hyphae formation in *Candida albicans* [277], but much higher concentrations were required to inhibit the growth of unicellular fungus *Saccharomyces cerevisiae* [278,279].

Some studies failed to detect inhibition of fungi by ZEN because the concentrations used (0.5 and 3.0 µg/mL) were too low [280]. Note that ZEN concentrations in cultures of *F. graminearum* can reach up to four orders of magnitude higher levels (Section 2.2).

#### 8.1.2. Stimulation of ZEN Production by Competing Fungi

Competition with other fungi was shown to stimulate ZEN production only in a few publications, while many antagonists have been reported to reduce ZEN levels. The problem with the latter results is that ZEN concentrations were not normalized in most of these studies. Antagonists and competitors suppress *Fusarium* growth, even if solely through competition for nutrients. Therefore, the concentration of ZEN measured in mixed cultures may drop, as reported in many studies (e.g., [90,281]), even if the antagonist stimulated ZEN production. In such experiments, it is necessary to relate ZEN concentration to the biomass of its producer, as Milles et al. explained [282]. An additional complication is that some antagonists may destroy ZEN by enzymatic hydrolysis. ZEN concentrations in mixed cultures with these organisms rapidly drop to zero (e.g., [283]).

Regardless of these complications, stimulation of ZEN production by competing microbes has been demonstrated. Co-cultivation of *F. graminearum* with *Penicillium verrucosum* and *Aspergillus ochraceus* increased the concentration of ZEN in the medium 4- and 5-fold, respectively, while the productivity in ZEN amount per unit of fungal biomass in co-cultures with *Penicillium verrucosum* increased 25-fold [282]. Co-cultivation with *Alternaria tenuissima* [284] and *Alternaria alternata* [284,285] and co-cultivation of *F. culmorum* with *Alternaria tenuissima* [286] also stimulated ZEN production. Examination of the effect of *C. rosea* on ZEN production by *F. graminearum* is masked by the activity of ZEN lactonase (Section 8.1.4). Therefore, N. Kirsch used a *C. rosea* strain with the disrupted ZEN lactonase gene for these experiments [287]. In all nine *F. graminearum* strains (three wild-type strains and six trichothecene- and aurofusarin-deficient mutants), ZEN concentration normalized to *F. graminearum* biomass increased substantially (mostly more than 10-fold) by co-incubation with *C. rosea*.

#### 8.1.3. Stimulation of ZEN Production by Competing Fungi Overlooked Due to Lack of Normalization

Velluti et al. [288,289] have reported no significant effect of co-cultivation with *F. verticillioides* and *F. proliferatum* on ZEN production by *F. graminearum*, though they observed an increase in the concentration of ZEN in the presence of the two competitors under certain conditions ([289], see also data for 15 °C in Figure 3 in [288]). ZEN levels were not normalized to *F. graminearum* biomass, which is known to be suppressed by co-cultivation with *F. verticillioides* producing fumonisins [290]. Suppression of *F. graminearum* growth by *F. verticillioides* remained unnoticed in these reports because spore production instead of fungal biomass was recorded as a proxy for fungal growth. Because most ZEN levels in mixed cultures remained unchanged while the growth of *F. graminearum* was presumably suppressed, we can assume that the production of ZEN increased.

Hoffmann et al. [291] studied interactions between *F. graminearum* with *Alternaria tenuissima* and *Pseudomonas simiae* in wheat ears. Both *F. graminearum* biomass and ZEN levels were reduced in all combinations of co-inoculation with *A. tenuissima*, but ZEN production three weeks after inoculation in the sequence *Alternaria*-*Fusarium*-*Fusarium* was actually stimulated as compared to control-*Fusarium*-*Fusarium*, though the effect remained unnoticed due to the lack of normalization. The biomass of *F. graminearum* was reduced by about four times, while the level of ZEN was reduced by about half (estimates from bar graphs in Figures 3 and 5 in [291]), indicating that ZEN production in wheat ears pre-inoculated with *Alternaria* approximately doubled.

#### 8.1.4. Degradation or Detoxification of ZEN by Fungi That Parasitize ZEN Producers

The first reported biological degradation of ZEN was the hydrolysis by *Gliocladium roseum* (now *Clonostachys rosea*), which was described in 1988 by then PhD student El-Sharkawy and his supervisor Abul-Hajj [292]. Their motivation was to find a microbial model of mammalian metabolism of ZEN. As strange as it may seem today, microbial models of animal metabolism were popular from the 1970s to the 1990s. Their *C. rosea* strain hydrolyzed the lactone bond of ZEN. The authors speculated that the same reaction may take place in mammals treated with Ralgro, which, as we know today, is not the case. The potential of the discovery for practical applications, let alone its implications for the biological function of ZEN, was not recognized. The publication received its first relevant citation a decade later in a review of biological detoxification of fungal toxins [293], but implications for the ecological function of ZEN still remained unnoticed.

In the 2000s, the enzymatic detoxification of mycotoxins became an attractive strategy, and two laboratories embarked on cloning the gene responsible for ZEN lactonase synthesis in *C. rosea*. A research group at DuPont/Pioneer Hi-Bred used differential transcriptomics, relying on the fact that ZEN lactonase in *C. rosea* was induced by α-ZEL but not β-ZEL [294]. Two sets of *C. rosea* cultures were incubated with 10 µg/mL of α-ZEL and β-ZEL, RNA was extracted, and transcripts present in α-ZEL-induced cultures but absent from β-ZEL-induced cultures were identified using the GeneCalling technique of CuraGene [295], which was the state-of-the-art industrial transcriptomics at the time. The gene was cloned, and Pioneer Hi-Bred applied for a patent [296] with a priority date of 27 March 2001. A U.S. patent [297] was issued in 2004, and patents in Canada, Mexico, Brazil, and other countries followed.

At the same time, Kimura’s laboratory in RIKEN, Japan, screened a collection of microorganisms for degradation activities towards ZEN and identified a strain of *C. rosea* that hydrolyzed the lactone bond of ZEN [298]. They purified the enzyme, obtained a partial amino acid sequence, designed PCR primers, and cloned the gene, which they designated *zhd101* [299]. With the gene at hand, they expressed the enzyme in *E. coli*, *S. cerevisiae*, and rice [300] and constructed a codon-optimized variant for *S. cerevisiae* [301]. RIKEN applied for a patent under the Patent Cooperation Treaty [302], but the application was rejected due to prior art by Pioneer Hi-Bred.

*C. rosea* has been known since 1958 to be a destructive mycoparasite [303], and it is particularly effective in suppressing ZEN producers *F. culmorum* [304] and *F. graminearum* [305]. The inhibition of fungal growth by ZEN and degradation of ZEN by *C. rosea* together substantiate the hypothesis that ZEN is a defense metabolite of *Fusarium* spp., and that *C. rosea* overcomes this defense by destroying ZEN enzymatically.

Genome mining for lactonase genes homologous to *zhd101* reveals the presence of such genes in the genomes of several mycoparasitic fungi, and laboratory experiments confirmed that *Trichoderma aggressivum* and *Clonostachys catenulatum* hydrolyze ZEN [306]. A ZEN lactonase gene from *T. aggressivum* has been cloned and its product characterized [307]. Incubation of *Trichoderma* spp. and *Clonostachys* spp. with autoclaved cultures of *Fusarium* spp. containing ZEN revealed that several *Trichoderma* spp. and *Clonostachys* spp. degraded ZEN, including *T. atroviride* and *T. lengibrachiatum* [283]. The ability of *T. atroviride* and *T. viride* to degrade ZEN was also demonstrated in pure cultures (Figure 6 in [308]). Both fungi parasitized *F. culmorum*, and they efficiently suppressed *F. culmorum* growth in dual cultures (Figure 11 in [308]). A ZEN lactonase gene from *T. aggressivum* has been cloned and its product characterized [307].

Apart from the hydrolysis of the lactone bond, enzymatic activities abolishing the biological activities of ZEN by modifications of its structure, referred to as detoxification, have been reported in many fungal species. Glycosylation of ZEN by many fungal species has been reported [309,310,311,312]; other detoxification activities are listed in many reviews, for instance by McCormick [313].

### 8.2. Proof by Gene Disruption Experiments

While randomly selected fungi and fungus-like stramenopiles were inhibited by ZEN at 10 µg/mL, *C. rosea*, which hydrolyzes ZEN enzymatically, was fully resistant to a twofold higher concentration [184]. Disruption of the gene encoding ZEN lactonase rendered *C. rosea* susceptible to ZEN [184]. The authors inferred that the results are consistent with the hypothesis that RALs, exemplified by ZEN, are interference competition agents suppressing the growth of competing fungi.

The experiment has often been cited as support for the biological function of ZEN in protecting *F. graminearum* against mycoparasites (for instance, most recently in [314,315,316]), which it clearly does, but it is merely an indirect support. Proof requires showing that ZEN enhances the fitness of *F. graminearum* under mycoparasitic attack. The first such proof has been obtained in a study of the role of ZEN lactonase in the biocontrol activity of *C. rosea* towards *F. graminearum* [317]. The study focused on the biological function of ZEN lactonase, but it also provided data on ZEN-producing *F. graminearum* and a gene disruption mutant not producing ZEN in interaction with a wild-type strain of *C. rosea* and its ZEN lactonase mutant. Challenge assay on agar plates and inoculation of wheat grains showed that the competitive fitness of *F. graminearum* strains not producing ZEN and/or their defense against mycoparasitism was reduced (Suppl. Figures 2 and 3 in [317]). N. Kirsch reported quantitative results in a similar in vitro system in her thesis [287]. She co-inoculated a *F. graminearum* strain producing ZEN or one of two ZEN-nonproducing mutants with *C. rosea* producing ZEN lactonase or a lactonase-deficient mutant on broken maize kernels. After 21 d, she quantified the biomass of both fungi by real-time PCR. In co-cultures of ZEN-producing *F. graminearum* with lactonase-deficient *C. rosea*, over 90% of fungal DNA was *F. graminearum* DNA (Figure 5). In co-cultures of ZEN-nonproducing mutants of *F. graminearum* with lactonase-deficient *C. rosea*, over 90% of fungal DNA was *C. rosea* DNA. In co-cultures with the *C. rosea* strain producing ZEN lactonase, essentially all fungal DNA in all mixed cultures after 21 d was *C. rosea* DNA.

The effect of the loss of ZEN synthesis on the competitive fitness of *F. graminearum*, shown in the left part of Figure 5, is very strong. The reason was that the cultures were inoculated with spore mixtures, ensuring close contact between the mycelia of both species within the entire colonized substrate. Traditional challenge assays on agar plates, which work with point-inoculated cultures growing on the agar surface, may miss fungistatic effects of metabolites that become diluted by diffusion into agar before they reach the target. This has been nicely demonstrated for the effect of fumonisins production on the competitive fitness of *F. verticillioides*: The loss of fumonisin production was not detectable by a traditional challenge assay on agar, while cultures on grains inoculated with a mixture of spores revealed significant and strong effects [290].

The experiments described above firmly established that ZEN is a defense compound protecting *F. graminearum* from the mycoparasite *C. rosea*. The function of ZEN in interference competition with non-parasitic competitors has not been rigorously proven, but it can be convincingly inferred from the facts that ZEN inhibits the growth of non-parasitic ascomycetes and *C. rosea* with disrupted ZEN-lactonase to a similar extent, and likely due to the same mechanism. Rigorous proof would require monitoring species-specific fungal biomass of ZEN-producing *Fusarium* sp. and ZEN-nonproducing mutants growing in mixed cultures together with fungal competitors. We expect that the results will resemble those in Figure 5.

### 8.3. Cautionary Note About the Use of ZEN and ZEN Mutants as Research Tools

Unnoticed physiological effects due to ZEN’s inhibition of heat shock proteins and/or MAP kinases, or the absence of these effects in ZEN-nonproducing mutants, may lead to incorrect conclusions in experiments that disregard the potential role of ZEN in the system. We provide two examples. The first example is the use of a ZEN-inducible promoter for genetic studies in *F. graminearum* [318]. The effects of ZEN on HSP90 and MAP kinases could lead to misinterpretation of experiments in which ZEN was used as an inducer. This would be particularly critical in a study of the function of *HSP90*; fortunately, the authors used estradiol instead of ZEN or β-ZEL as an inducer because of its lower price [319].

The second example is a study of the effect of mitochondrial viruses on mycotoxin production by *F. graminearum* and food choice of predators, on which we participated [315]. *F. graminearum* strain with disrupted ZEN synthesis was used as a recipient for the transmission of the virus because of its resistance to hygromycin facilitated counterselection of the donor. Comparisons between the virus-free and the transfected strains were not affected, but important differences between the donor and the acceptor, such as their ability to support the replication of the virus, may have been influenced by processes modulated by ZEN accumulating in the donor.

## 9. Biological Function of Fungal Resorcylic Acid (RALs) and Dihydroxyphenylacetic Acid (DHPLs) Macrolactones

### 9.1. The Hypothesis

Macrocyclic lactones similar to ZEN have been isolated from many fungal species. They are derived from resorcylic and dihydroxyphenylacetic acids, and we will refer to them collectively as macrolactones. They are widespread among fungi; a review from 2015 counted 190 of them [320], and many are produced by several fungal species. Antifungal activities have been reported for many macrolactones, and they may exist, though unreported, in others. Lactonase hydrolases have been found in many fungi, too. Together with the proofs of the functions of ZEN in *F. graminearum* and the proof of the function of ZEN lactonase in *C. rosea*, these facts support the generalization from ZEN to other macrolactones. Thus, we hypothesize that the ecological function of RALs and DHPLs in fungi is defense against mycoparasites and competitors, which we collectively call antagonists. The hypothesis builds upon and generalizes previous suggestions that monorden may play a key role in the parasitism of *Monocillium nordinii* on pine-pine gall rust fungus [321], that other mycoparasites may benefit from the production of metabolites of the same class [322], and that Hsp90 chaperone may be a prominent target in the establishment of interactions of RAL-producing fungi with other organisms [323].

RALs and DHPLs require different synthetic strategies in the lab because in RALs the acyl group of the lactone is attached directly to the aromatic ring, but in DHPLs it is separated from the aromatic ring by a benzylic CH_2_. Because members of both classes of macrolactones (Figure 6) have been shown to be produced by *Acremonium zeae* (now *Sarocladium zeae*) [324], but the genome of the producer (GeneBank GCA_048934535.1) contains only one homologue of *PKS4* and one homologue of *PKS13*, the same PKSs apparently catalyze the synthesis of both structures. This suggests a closer relationship between RALs and DHPLs than previously recognized.

### 9.2. Fungal Homologues of Polyketide Synthases Responsible for ZEN Synthesis

In 2016, Dawidziuk, Koczyk, and Popiel inferred that the synthesis of fungal RALs and DHPLs is likely of ancestral origin, predating the split between *Ascomycetes* and *Basidiomycetes* [280]. Their argument was partly based on a discovery of RALs in the *Basidiomycetes* fungus *Sporotrichum laxum* [325]. We may add to their reasoning that although the RALs produced by *Sporotrichum laxum* (spirolaxine, sporotricale, and sporotricale methylether) are not macrolactones, other products (phanerosporic acid and a new compound No. 10) resemble the last intermediate of ZEN synthesis just before cyclization to β-ZEL (Figure 3). The compound No. 10 was described in 1997 in a Japanese patent [326]. They specified *Chrysosporium pruinosum* as the source, which is also a basidiomycete. Phanerosporic acid was first isolated from cultures of another basidiomycete, *Phanerochaete chrysosporium* [327]. Although phanerosporic acid is not a lactone, it shares antifungal activity with ZEN and other RALs. Interestingly, it also stimulated plant growth [327] similarly to ZEN (Table 3).

Since Dawidziuk et al.’s work [280], many fungal genomes have been added to databases. We searched for homologues of PKS13 and PKS4 in virtual proteomes of fungi and present the results in Figure 7 and Figure 8. For most of the homologues found, we were able to identify the likely metabolic product. Two of them are not macrolactones, but both cladosporin [328,329] and cytosporones [330] exhibit strong antifungal activities. Table 1 shows the acc. Nos. of the PKSs predicted to be responsible for the synthesis of cladosporin, hypothemycin, and radicicol in 11 fungal species.

### 9.3. Antifungal Activity of RALs and DHPLs

The first fungal RAL was isolated in 1953 from *Monosporium bonorden* due to its fungicidal activity [331]. In 1964, the structure of the metabolite was established, and it was designated monorden [332]. In the same year, antifungal metabolite radicicol was isolated from *Cylindrocarpon radicicola*, and its structure was found to be identical to monorden [333]. The same metabolite was later isolated from many other species, for instance, *Colletotrichum graminicola* [334], *Chaetomium chiversii* [238], *Penicillium luteo-aurantium* [335], *Humicola fuscoatra* [322], *Neocosmospora tenuicristata* [336], and *Niesslia endophytica* [337].

Producers of radicicol comprise fungi of all lifestyles: saprophytes, endophytes, plant pathogens, and mycoparasites. Table 6 presents selective macrolactones with well-documented antifungal properties. Fungicidal activities of RALs and DHPLs are so strong that they have been considered for the control of Dutch elm disease [321]. Radicicol was recently suggested as a lead structure for the development of fungicides for the control of plant diseases [338].

### 9.4. Conservation of Molecular Targets Hinders Adaptation of Antagonists

Cytotoxic activity of curvularin and ZEN towards HeLa cells has been reported already in the 1970s [339], but the molecular target was unknown. Major medical interest in RALs was sparked by the discovery that radicicol reversed the malignant phenotype of oncogene-transformed fibroblasts [340]. The effect was originally attributed to the inhibition of certain protein kinases, but the main molecular target of radicicol was found to be the heat shock protein HSP90, in which radicicol blocks the ATP/ADP-binding site [235,341]. As this part of HSP90 is highly conserved (in *E. coli*, HSP90 is designated HtpG), radicicol inhibits molecular chaperones of this family from bacteria through yeast to mammals [342]. Radicicol and its dechlorinated derivative monocillin I were also shown to bind to and inhibit plant HSP90 [343]. The resorcinol moiety (2,4-dihydroxy-6-alkenylbenzoic acid) is important for binding of radicicol to the chaperone [344].

**Table 6 toxins-17-00226-t006:** Fungal RALs and DHPLs with documented antifungal properties.

Name	Chemical Structure(Example) ^1^	Fungal Producers (Incomplete List)	Ref. toIsolation	Ref. to Anti-Fungal Activity
Cryptosporiopsin A	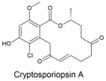	*Cryptosporiopsis* sp.	[345]	[345]
Curvularindehydrocurvularin	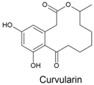	*Curvularia aeria*, *Alternaria*, *Penicillium* sp., *Cochliobolus spicifer*, *Alternaria longipes*	[346,347,348,349]	[350]
Hypothemycin	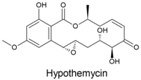	*Hypomyces trichothecoides*, *Hypomyces subiculosus*, *Podospora* sp.	[351,352,353]	[354]
Lasiodiplodin	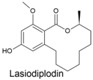	*Lasiodiplodia theobromae*, *L. pseudotheobromae*	[355]	[355]
Monocillin VI and 4′-hydroxymonocillin IV	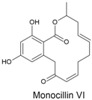	*Paecilomyces*	[356,357]	[358]
Queenslandon	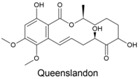	*Chrysosporium queenslandicum*	[359]	[359]
Radicicol (monorden)	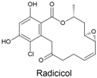	Many ^2^	Many ^2^	[321,322,338,360]

^1^ One of the compounds listed in the 1st column is shown, but all were shown to possess antifungal properties. ^2^ Eight producers of radicicol, together with references, are listed at the beginning of Section 9.3.

The second prominent target for RALs is mitogen-activated protein kinases (MAP kinases) [239,361]. The affinity of radicicol itself to kinases is low, but derivatives such as radicicol A and other RALs bind to and inhibit MAP kinases [235]. Similarly to HSP90, the binding site is the ATP-binding pocket of MAP kinases. Because activation of MAP kinases is involved in the development of different kinds of cancer, fungal RALs have been studied as promising leads for antineoplastic drugs [362].

Chaperones support the refolding of many proteins, and MAP kinases are involved in several transduction pathways. Interfering with these functions impairs vital processes in the cell. Conservation of the targets, which in both proteins are the ATP-binding pockets, likely hinders antagonists’ adaptation by target modification. We suggest that this is the reason why this chemical defense works despite eons of coevolution between the producers of macrolactones and their antagonists. This is also the likely reason why many parasites of *Fusarium* spp. developed or acquired the ability to enzymatically degrade or detoxify ZEN.

### 9.5. Self-Protection of Macrolactone Producers Is Inefficient

The same reason, however, hinders the self-defense of macrolactone producers. Functional constraints due to conserved targets restrict target modifications, and active transport out of the cell only removes a freely diffusing molecule after it reaches the transporter, which limits its efficiency at low concentrations. Fungi can synthesize toxic metabolites in vesicles that discharge their content to the extracellular space by exocytosis [363], which overcomes the limitations of self-defense, but there is no indication that macrolactones are synthesized in vesicles.

Data support the expectation that the self-defense of macrolactone producers is inefficient. Kang et al. [364] developed a selective medium for *Cylindrocarpon destructans*, which produces radicicol, by supplementing the growth medium with 50 µg/mL radicicol. Inspection of Figure 2 in [364] shows that radicicol at this concentration reduced the growth of *C. destructans* by about 50%.

Investigation of the HSP90 from the radicicol producer *Humicola fuscoatra* led to similar results [365]. The affinity of radicicol to HSP90 from *H. fuscoatra* was lower than to other HSP90 proteins, which the authors explained by two substitutions in the ATP binding site, but yeast expressing the protein was only moderately resistant to radicicol. The most resistant form of HSP90 was obtained by introducing the substitutions from HSP90 of *H. fuscoatra* into the yeast protein, but yeast expressing the protein was still partially inhibited by 60 µM radicicol.

The situation with *F. graminearum* and ZEN is similar. According to a conference abstract from 2011, HSP90 from *F. graminearum* is resistant to ZEN [366]. This resistance, however, is limited because the presence of 50 µg/mL ZEN in liquid cultures inhibited *F. graminearum* growth (our unpublished results), though *F. graminearum* rapidly converts external ZEN to ZEN-4-sulfate [366] (Note: production of ZEN sulfate by *F. graminearum* has been known since 1991 [367]).

### 9.6. Fitness Costs and Control of Macrolactone Synthesis

Production of macrolactones reduces the fitness of the producers by inhibiting their growth due to inefficient self-defense and by incurring metabolic costs. Inefficient self-protection has been documented in the previous section. Metabolic costs are likely to be substantial, too, because macrolactones accumulate in cultures of their producers in large amounts. Regarding ZEN, this is described in Section 2.2. Remarkably high concentrations of radicicol, pochonins, and monocillins in fungal extracts were pointed out by Piper and Mills [323]. To maintain macrolactone production, fitness reduction must be compensated by a larger fitness gain. The largest biotic threats to fungi are predators, parasites, and competitors. The protective effects of *F. graminearum* against mycoparasites by ZEN have been proven, and protection against competitors is likely because antifungal effects of ZEN are not species-specific, but ZEN has not protected *F. graminearum* from grazing by springtails and isopods (Figure 9 and further results in [368]). Furthermore, ZEN synthesis was suppressed by predation (see next section). Similarly, monordens and monocillins have not deterred corn earworm larvae from feeding and exhibited low toxicity to them [322]. Thus, fitness reduction due to metabolic costs of macrolactone synthesis and self-inhibition must be compensated by the fitness gain due to protection against mycoparasites and/or competitors.

Exposure of fungi to mycoparasites and competitors, which we will collectively call antagonists, varies with time and space. To maximize their fitness, fungi must tune their defense to match not just current but also future exposure because the onset of transcription and the protective effect are separated by a time gap. Chemical defense should also match exposure to antagonists that are likely to occur in the future. How can fungi achieve it?

Maximization of fitness has driven the development of gene regulatory networks, including those that control defense chemicals. Fungal life histories involve scenarios with different requirements for chemical defense. Each scenario is endowed with a different pattern of environmental signals. We believe that the perception and processing of these signals enables fungi to launch defense compound synthesis with optimal timing and intensity, and to turn the synthesis off when appropriate. The complexity of regulatory networks governing chemical defense must match the complexity of life history scenarios. That ZEN synthesis in *F. graminearum* is controlled by several regulatory networks interacting with each other (Section 3) is in line with this reasoning.

### 9.7. Regulatory Networks and Life History Scenarios

Except for ZEN, ecological and physiological data on macrolactone synthesis in fungi are scarce. Therefore, examples of life history scenarios mainly rely on ZEN production by *F. graminearum*. Apart from ZEN, fumonisins are the only metabolites of *Fusarium* whose role in interference competition has been proven [290]. Similarly to ZEN, production of fumonisins is controlled by multiple gene regulatory circuits, including a pathway-specific transcription factor [369] of factors with multiple binding sites [370], epigenetic control [371,372], AreA-dependent nitrogen repression [373], and mRNA methylation [374].

The first life history scenario examines colonization of nutrient-rich seeds. Mycoparasites and competitors are absent, but defense compounds may accumulate, as suggested for fumonisins accumulating in maize kernels infected with *F. verticillioides*. Fumonisins are virulence factors in rice [375] and may play a role in the infection of maize seedlings under certain conditions [376,377], but they are not involved in the infection of maize ears by *F. verticillioides* [378,379,380]. In spite of that, infected maize kernels accumulate substantial amounts of fumonisins. Sherif et al. [290] hypothesized that fumonisins in these kernels suppress competitors of *F. verticillioides* after the grains fall on the ground. The observations that fumonisins are produced mainly in late stages of seed development [381] and that their synthesis is enhanced by co-infection with *F. graminearum* (Figure 9 in [290]) support this hypothesis. How has the gene regulatory network evolved? One can envision that through its evolution, *F. verticillioides* repeatedly colonized starch- and oil-rich seeds under declining water activity and absence of microbial metabolites. Strains that responded to these conditions by moderate production of fumonisins may have gained maximum fitness; strains that produced less fumonisins may have been outcompeted by fungi after the seeds have fallen on the ground; and strains that produced higher than optimal amounts of fumonisins may not have built up enough biomass due to metabolic costs and self-inhibition. We hypothesize that the scenario of pre-emptive protection of nutrient-rich substrates also applies to *Fusarium* spp. producing ZEN and to plant pathogens producing other RALs or DHPLs such as *Cylindrocarpon radicicola*, *Ilyonectria mors-panacis*, *Colletotrichum graminicola*, and *Lasiodiplodia theobromae*. We know that ZEN accumulates in infected grains at late stages of infection [250,251], and our hypothesis predicts that the accumulation of other RALs or DHPLs exhibits similar kinetics.

The second scenario considers a macrolactone producer facing competition from fungi colonizing the same substrate. We may expect that the presence of competitors triggers defense compound production, though experimental confirmation is only available for ZEN (Section 8.1.2). Different competitors likely differ in their ability to cope with defense metabolites, and producers of defense metabolites may distinguish among competitors or groups of competitors, similarly to mycoparasites that distinguish among prey species [382], even at a distance [383]. A regulatory network tuned by evolution to maximize the fitness of a defense compound producer is expected to process competitor-specific signals to launch the most effective defense strategy. Turning off the synthesis when it is not needed is also an important function of a regulatory network.

Finally, we consider a scenario with different threats occurring concomitantly, choosing *F. graminearum* as an example. Aurofusarin is the major—if not only—metabolite protecting *F. graminearum* from predators [368]. Both aurofusarin and ZEN are polyketides, and their syntheses compete for precursors [384]. Because the threat of predation is more severe than competition or parasitism, a strategy maximizing fungal fitness under predation pressure must maximize aurofusarin synthesis, which requires channeling all polyketide precursors into the aurofusarin pathway. RNAseq data generated in our study of the biological function of aurofusarin [368] confirmed this prediction by showing that ZEN synthesis was suppressed under predation (Table 7).

The key effect revealed by Table 7 is the suppression of *ZEB2*, which is strictly necessary for ZEN synthesis [59]. Note: Decreased ZEN concentration in cultures subjected to predation would not support the hypothesis that ZEN synthesis is suppressed by a gene regulatory network because aurofusarin synthesis stimulated by predation [368] is expected to deplete biosynthetic resources for ZEN, as observed in aurofusarin-overproducing mutants [384].

## 10. Research Challenges

### 10.1. Asking the Right Questions

When the synthesis of a new fungal special metabolite is elucidated today, gene disruption mutants are usually generated to reveal the function of individual genes in the pathway. With mutants in the key gene of the pathway at hand, which most often will be a terpene, polyketide, or nonribosomal peptide synthase, the biological function of the metabolite in chemical defense can be investigated. This is rarely done, even in situations when the hypothesis is compelling and should be immediately apparent. One of the reasons may be a bias within the research field. In a laboratory studying plant–pathogen interactions, mutants with disrupted synthesis of a new specialized metabolite are likely to be subjected to an infection assay. If the result is negative, the experiments will be repeated, the infection conditions will be tweaked, and other host plants will be tested. If everything fails, the study concludes with negative results.

It is only recently that plant pathologists began appreciating the role of the saprophytic phase in the life cycle of necrotrophic pathogens, including the role of mycotoxins in the formation of inoculum for plant infection [385]. Biotic challenges faced by pathogens in their life cycle, except for the defense responses of the host, are seldom considered. Biotrophic pathogens are subject to mycoparasitism. Necrotrophic pathogens spend most of their life cycle saprophytically, facing threats of predation, competition, and mycoparasitism. These permanent threats likely dominated selection forces driving the development of specialized metabolite synthesis in necrotrophic fungi to a larger extent than pathogens’ interactions with host plants.

### 10.2. Example: Fusaristatin A in F. pseudograminearum

Fusaristatin A is a hybrid polyketide-nonribosomal peptide isolated for the first time from an endophytic strain of *Fusarium* [386]. The metabolite is produced by many species of *Fusarium* and other fungal genera, but it was believed not to be produced by *F. pseudograminearum* because the corresponding biosynthetic cluster was missing from the *F. pseudograminearum* genome [387]. Five years later, *F. pseudograminearum* strains possessing the biosynthetic cluster and producing fusaristatin A have been found in Western Australia, while isolates from the other states of Australia did not produce fusaristatin [388]. Remarkably, strains with disrupted fusaristatin synthesis grew faster and infected wheat more severely than fusaristatin A producers [389]. Thus, as the authors noted, fusaristatin synthesis negatively impacted both the saprophytic and the pathogenic stages of the life cycle. Populations that have lost fusaristatin coexist with fusaristatin producers, which the authors found perplexing. We suggest, with a degree of certainty, that the solution to the puzzle is that fusaristatin A protects *F. pseudograminearum* against predation. We hypothesize that the fitness gain in areas with strong predation pressure outweighs the fitness depression due to costs of synthesis and/or self-toxicity, and the synthesis is maintained by positive selection. Fitness gain in areas with low predation pressure is lower than the costs of synthesis, and the synthesis is counterselected. Data shown in Figure 10 support this hypothesis: expression of all genes of the fusaristatin gene cluster was stimulated by springtail grazing.

### 10.3. Avoiding Pitfalls

An often-encountered pitfall in research on specialized metabolites is a lack of normalization of metabolite concentration to the biomass of the producer. Whether mycotoxin content of field grains or metabolomes of laboratory cultures, normalization is indispensable for any inference drawn from quantitative data. Plant breeders have reported positive correlations between DON levels in field grains and disease score, and often inferred that positive correlations corroborate the role of DON in virulence. This is incorrect. Isolates with higher aggressiveness grow more biomass in planta, and they cause more severe symptoms. Any metabolite of a necrotrophic pathogen produced during the infection is expected to correlate with the biomass of the pathogen and thus with disease symptoms. To infer the role of a metabolite in virulence, its concentration in plant tissue must be normalized to fungal biomass. This has rarely been done, which is why we could cite only a few such correlations in Table 5.

The opposite situation occurs in laboratory studies of the effect of competition on the production of fungal metabolites. If the concentration is not normalized, stimulation of synthesis by competition or predation may become masked by the reduction of fungal biomass. Examples are shown in Section 8.1.3.

A second common pitfall is the use of traditional challenge assays on agar plates in studies of fungal interactions. On agar surface, interacting cultures do not have direct contact before their colonies meet, and their chemical interaction via secreted metabolites is weakened by diffusion of the metabolites into agar. The biomass of a fungal colony on an agar surface is much smaller than the volume of the agar medium, which is 95% water. Secreted metabolites undergo strong dilution before they reach the interacting partner. Competitive interactions between saprophytic or necrotrophic fungi should be investigated by inoculating natural substrates such as grains with mixtures of spores or mycelial fragments, which results in truly mixed cultures consisting of strains or species closely interacting within the entire substrate. Liquid cultures inoculated with mixtures of spores also guarantee close interaction within the entire culture, but secreted metabolites are strongly diluted, especially in early stages of the incubation, weakening their effect on the interacting partner. Another drawback of liquid cultures is that, except for aquatic fungi, they do not well simulate the natural environment of fungi.

Inadequate use of ELISA, which was a common pitfall in the past (Section 4.2), is unlikely to occur anymore.

### 10.4. Access to Gene Disruption Mutants

Pathway disruption mutants are the key tools in research on specialized metabolites. Methods for the generation of such mutants are now widely available, but it is a laborious procedure requiring special skills and equipment. Therefore, strains carrying gene disruptions should be preserved and made available for future research. The value of these strains grows with each published study. Typically, strains are kept in local culture collections of the laboratories where they have been generated, as long as the laboratories exist. Unfortunately, local collections are not permanent. For instance, a collection from which we obtained many gene disruption mutants of *F. graminearum* ceased to exist a few years ago. If we had not kept the strains, they would be lost to science.

We urge all colleagues to deposit their strains in curated collections that guarantee long-term availability such as CABI Genetic Resource Collection (GRC) in Egham, Surrey, UK; Fungal Genetic Stock Center at Kansas State University in Manhattan (KS), USA; Westerdijk Fungal & Yeast Collection (formerly CBS) in Utrecht, The Netherlands; or DSMZ—German Collection of Microorganisms and Cell Cultures in Braunschweig, Germany. It is important to deposit entire sets, with wild-types, ectopic insertions, and complemented strains or several independent disruption mutants. This is necessary because strains with deleted or disrupted targets usually contain selection markers, and they may also carry unknown mutations or genome rearrangements inadvertently introduced during the transformation, which may affect their behavior in assays used to examine the function of target genes. The chemical phenotype of strains with ectopic insertions of resistance cassettes may be effected in unexpectable ways; for instance, enzymes responsible for antibiotic resistance may phosphorylate or acetylate fungal metabolites resembling their substrates. Additional independent strains with independent disruptions of the target gene or complemented strains are also needed to exclude effects of inadvertent mutations. Rapid access to sets of strains comprising independent disruption mutants or complemented strains and ectopic insertions enables laboratories in ecological chemistry lacking molecular skills to help us understand why fungi produce specialized metabolites.

## Figures and Tables

**Figure 1 toxins-17-00226-f001:**
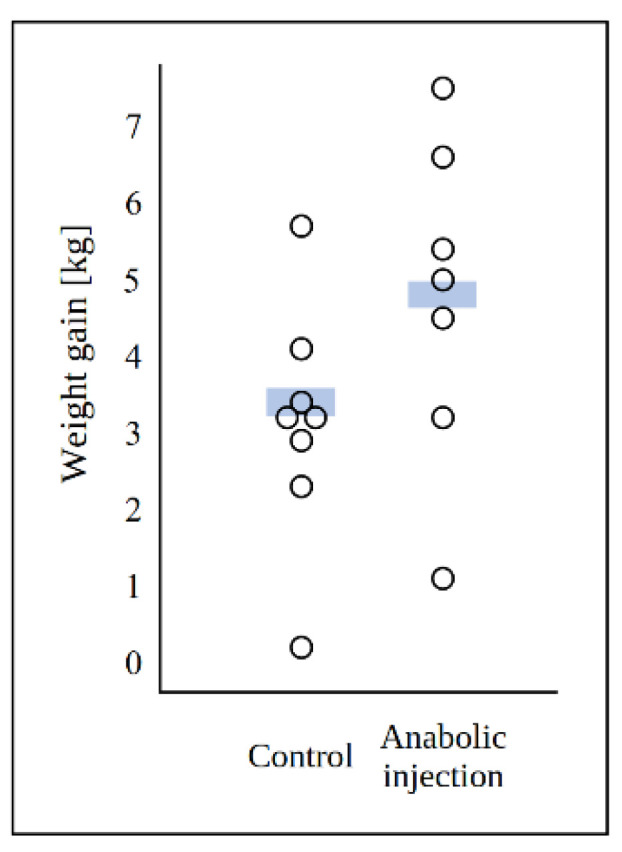
The first report of the anabolic effects of ZEN. The weight gain in lambs injected daily with “anabolic substance” purified from *F. graminearum* after 28 d is shown. The graph was constructed using data from a U.S. patent [8], which resulted from a series of continued-in-part applications filed since December 1960. Circles show the weight gain of individual animals; blue rectangles show the means.

**Figure 2 toxins-17-00226-f002:**
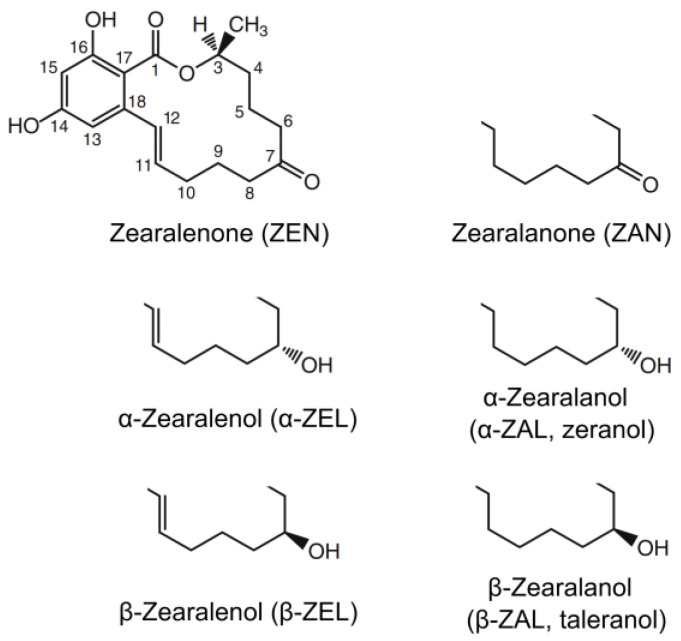
Structures, standard acronyms, and recommended numbering of carbon atoms of zearalenone and its major derivatives. For the derivatives, only fragments covering atoms 6 to 12 are drawn.

**Figure 3 toxins-17-00226-f003:**
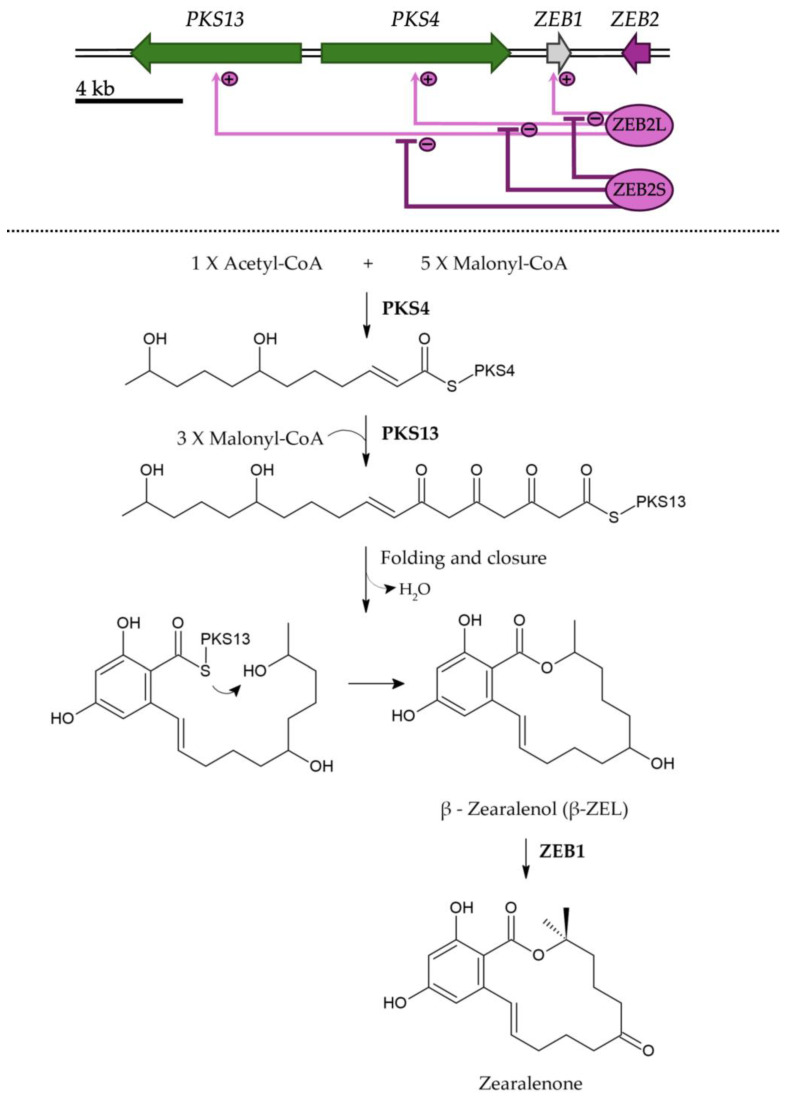
ZEN biosynthetic pathway in *F. graminearum* and its control by activator ZEB2L and suppressor ZEB2S. In the upper part of the figure, arrows with plus indicate induction, and blunt lines with minus indicate suppression.

**Figure 4 toxins-17-00226-f004:**
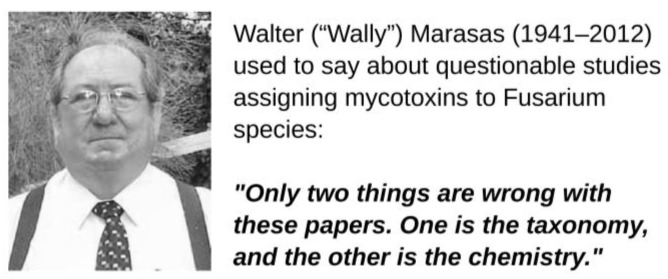
Walter Marasas’s comment about the wrong assignments of *Fusarium* mycotoxins to the producing species. Modified photo from [154] was used in line with the Creative Commons Attribution 4.0 International License [155].

**Figure 5 toxins-17-00226-f005:**
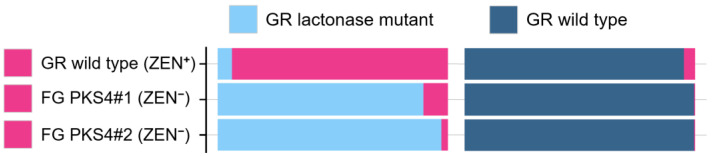
Relative amount of species-specific DNA in co-cultures of *F. graminearum* (GR) and *C. rosea* (CR). Dual cultures of *F. graminearum* strain 2311 producing ZEN and two mutants not producing ZEN with *C. rosea* strains DSM 113708, which is a ZEN lactonase disruption mutant of DSM 62726 (**left**), and DSM 62726, which is a wild-type strain producing ZEN lactonase (**right**), were grown on maize kernels for 21 d at 24 °C. Total DNA was extracted, and species-specific DNA was quantified by real-time PCR. The results are presented as fractions of total fungal DNA. Figure from the thesis of M. Kirsch [287] was modified and reprinted with permission of the author.

**Figure 6 toxins-17-00226-f006:**
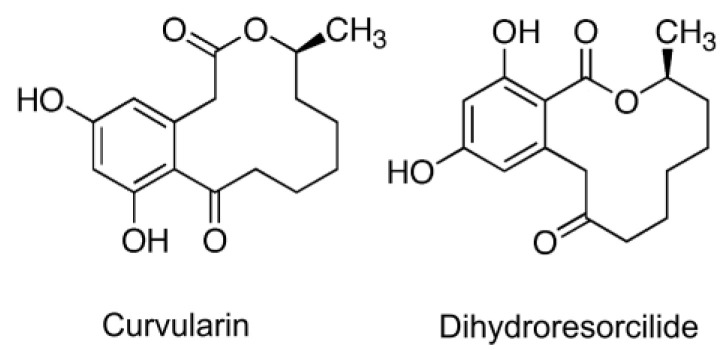
*Sarocladium zeae* produces both DHPLs (**left**) and RALs (**right**).

**Figure 7 toxins-17-00226-f007:**
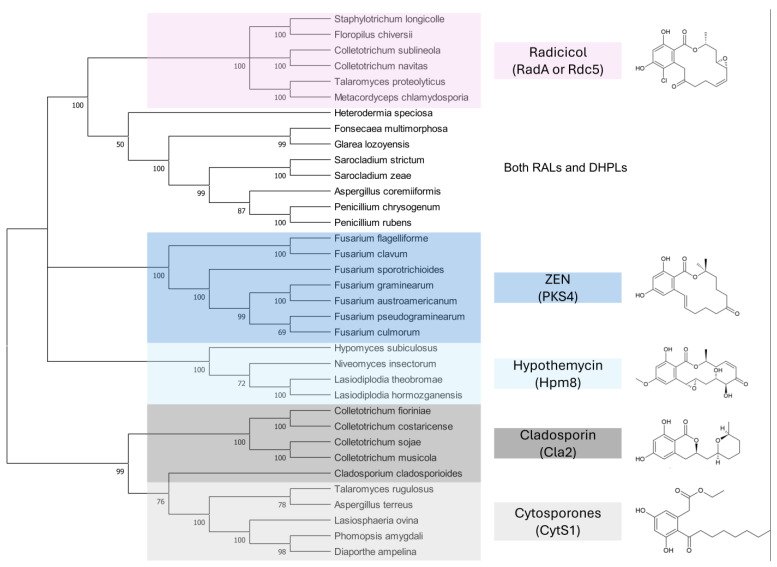
Homologues of PKS4 from *F. graminearum* in fungal genomes. The dendrogram was constructed using Maximum Likelihood based on BLASTP version Blast+2.16.0 for transcribed genes (“virtual proteoms”) sharing more than 70% identity and more than 90% query cover with FGSG_12126 (PKS4) from *F. graminearum*. Bootstrap values (500 replicates) indicate branch support. Proteins from other fungal species are labeled accordingly.

**Figure 8 toxins-17-00226-f008:**
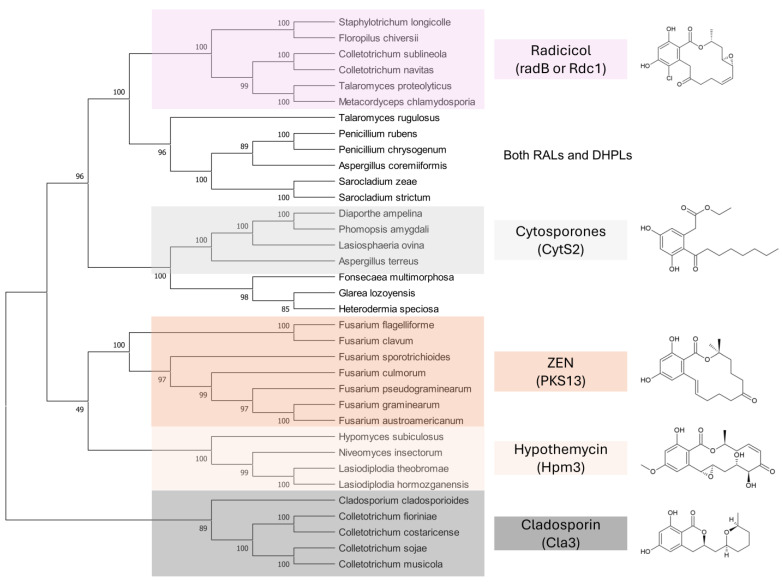
Homologues of PKS13 from *F. graminearum* in fungal genomes. The dendrogram was constructed using Maximum Likelihood based on BLASTP version Blast+2.16.0 for transcribed genes (“virtual proteoms”) sharing more than 70% identity and more than 90% query cover with FGSG_02395 (PKS13) from *F. graminearum*. Bootstrap values (500 replicates) indicate branch support. Proteins from other fungal species are labeled accordingly.

**Figure 9 toxins-17-00226-f009:**
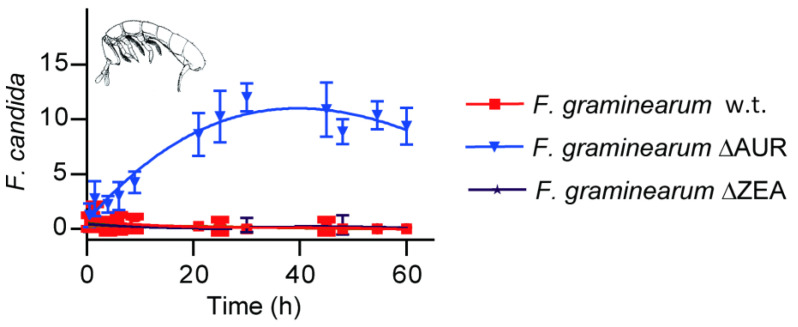
Preference of the springtail *Folsomia candida* for grazing on a *F. graminearum* strain producing aurofusarin and ZEN and its gene disruption mutants. Springtails were offered a choice between colonies of the wild-type (w.t.) strain, which was a producer of aurofusarin and ZEN, ∆AUR (a mutant with disrupted aurofusarin synthesis), and ∆ZEA (a mutant of the wild type with disrupted ZEN synthesis). The number of animals grazing on a particular strain at different time points is shown. Data for ∆ZEA (black line) coincide with and are partially covered by data for the w.t. Modified from Suppl. data in [368] shown in line with Creative Commons Attribution 4.0 International License [155].

**Figure 10 toxins-17-00226-f010:**
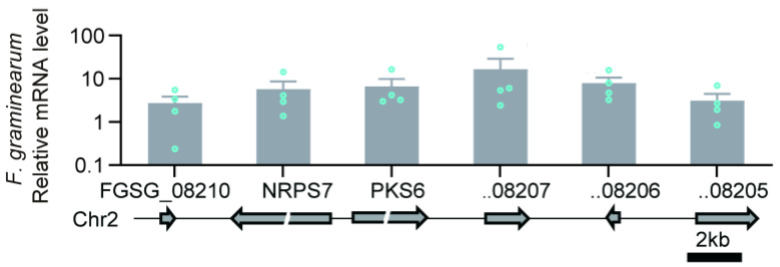
Springtails grazing stimulates expression of the gene cluster encoding fusaristatin A synthesis in *F. graminearum*. Data points show the change fold factors for grazing/control for four replicates. RNAseq data from [368] used in line with Creative Commons Attribution 4.0 International License [155].

**Table 2 toxins-17-00226-t002:** Homologues of *PKS4* and *PKS13* of *F. graminearum* in completely sequenced *Fusarium* genomes.

Species Name	Reference Genome Assembly	PKS13				PKS4			
		Identity (%) ^1^	Coverage (%) ^2^	Length (bp)	Accession No.	Identity (%) ^1^	Coverage (%) ^2^	Length (bp)	Accession No.
*Fusarium* *graminearum*	ASM24013 v3	100	100	6322	FGSG_02395	100	100	6876	FGSG_12126
*Fusarium * *austroamericanum*	ASM1765703v1	99	100	6323	JAGDVG010000009 1239521… 1235114	99	100	6875	JAGDVG010000009 1233581… 1226706
*Fusarium pseudo-graminearum*	FP7	95	100	6111	FPSE_06864	97	100	6875	NC_031951 7813499… 7806624
*Fusarium * *culmorum*	ASM1695235v1	95	100	6329	CP064750 7855407… 7849078	97	100	6875	CP064750 7863815… 7856940
*Fusarium * *sporotrichioides*	ASM1905464v1	93	98	6311	JAHKNS010000005 1036321… 1030010	94	100	6874	JAHKNS010000005 1028384… 1021510
*Fusarium * *flagelliforme*	Fuseq1	86	94	6149	XM_046132901	88	99	6869	NW_025763510 300333… 293464
*Fusarium clavum*	ASM4464674 v1	88	88	6317	JAVTNS010000027 434237… 427920	88	99	6871	JAVTNS010000027 426204… 419333
*Fusarium* *semitectum* (*F. incarnatum*)	ASM370943 v1	86	87	5959	RBJE01000009 347467… 341508	87	99	6866	RBJE01000009 356373… 349507
*Fusarium * *equiseti*	ASM331317 v1	87	82	5185	QOHM01000007 280762… 275577	88	99	6871	QOHM01000007 290435… 283564
*Fusarium * *verticillioides*	ASM14955 v1	—	—	—	—	—	—	—	—
*Fusarium * *oxysporum*	ASM1308505 v1	—	—	—	—	—	—	—	—
*Fusarium solani*	NDSU_Fsol_1.0	—	—	—	—	—	—	—	—

^1^ Only genes with >70% identity to the query are shown; ^2^ only genes with >50% query cover are shown.

**Table 4 toxins-17-00226-t004:** Research on ZEN as an endogenous hormone of plants.

Year	Plant	Note	Ref.
1986	Winter wheat and carrot	ZEN-like metabolite and its role in vernalization	[213]
1986	*Brassica campestris*	ZEN-like metabolite and its role in vernalization	[214]
1989	Winter wheat	Isolation of ZEN from shoot apices	[215]
1989	*Apium graveoleus*	Isolation of ZEN from wild celery	[216]
1990	Winter wheat and cotton	Variation of ZEN content during development	[217]
1990	Winter wheat	Variation of ZEN content during vernalization	[218]
1992	Wheat, *Lemna aequinoctialis*	ZEN appears to control plant development	[219]
1994	Winter wheat	Variation of ZEN content during development	[220]
1995	*Nicotiana tabacum*	ZEN appears to play a role in floral development	[221]
1996	-	Review	[222]
1997	Maize and wheat	ZEN conjugates with seed proteins	[223]
1998	*Cannabis sativa*	ZEN content in flower primordia	[224]
1999	Sweet corn	ZEN content in tassels	[225]
2000	*Nicotiana tabacum*	ZEN content correlated with flower development	[212]
2001	-	Review	[226]
2010	-	Review	[176]

**Table 5 toxins-17-00226-t005:** Support for the function of ZEN and DON as virulence factors (VFs) in wheat.

Examination or Experiment	Observation or Result Expected for VFs	ZEN	DON
Kinetics and spatial location of production during infection	Produced at early stages and at the front of infection	No [249,250,251]	Yes [252,253,254,255]
Treatment of host plants with purified metabolite	Causes similar symptoms to the infection	No (Table 3, [191,228])	Yes [256]
Comparison of field isolates	Production of a VF correlates with virulence	No ^1^ [257,258,259,260]	Yes ^2^ [261,262]
Biological function of similar metabolites	Similar metabolites have been proven to be VFs of their producers	No	Yes ^3^ [263,264,265]
Equip host plants with detoxification ability	Plants detoxifying VFs became more resistant to infection	?	Yes [266,267,268,269]
Gene inactivation	Disruption of biosynthesis reduced virulence	No ^4^ [58,149]	Yes [270,271,272,273]

^1^ ZEN levels in infected grains in [258] were not normalized, but three of the four most virulent isolates have not produced any ZEN. In [259], the more virulent strain produced less ZEN than the less virulent strain both in vitro and in planta. ^2^ Data in [261] were obtained on rye infected with *F. culmorum*, which is similar to wheat infection with *F. graminearum*. In [262], strains differing in DON production were generated by genetic engineering. ^3^ The metabolites are nivalenol, T-2 toxin, and diacetoxyscirpenol. ^4^ Additionally, [59] showed that disruption of ZEN synthesis in *F. graminearum* has not affected the infection of barley ears, which is similar to the infection of wheat ears.

**Table 7 toxins-17-00226-t007:** Effect of predation on the expression of ZEN cluster in *F. graminearum*.

Gene	Treatment	Expression ^1^	Fold Change	*p*-Value
*PKS13*	Control	0.042	0.69	NS
	Predation	0.029		
*PKS4*	Control	0.053	0.17	NS
	Predation	0.009		
*ZEB1*	Control	133	0.60	<0.001
	Predation	79		
*ZEB2*	Control	34	0.57	<0.05
	Predation	19		

^1^ Ratios of transcription rates under predation to controls. Suppl. data from [368] were used in line with Creative Commons Attribution 4.0 International License [155]. The units are FPKM (fragments per kb of transcript per million fragments mapped. NS: not significant. For details, see [368].

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
