# Peer review of "A Comprehensive Review of Hypotheses About the Biological Function of Zearalenone, and a New Hypothesis for the Function of Resorcylic and Dihydroxyphenylacetic Macrolactones in Fungi"

_toxins, 2025, doi:10.3390/toxins17050226_

Round 1

Reviewer 1 Report

Comments and Suggestions for Authors

The manuscript covers a wide range of topics, including the discovery, biosynthesis, hypotheses about biological functions, and ecological roles of ZEN and related compounds. It provides a thorough and detailed review of the biological functions of zearalenone and related fungal metabolites, with a strong emphasis on their role in fungal defense mechanisms. The review is well-structured, comprehensive, and supported by extensive literature. However, there are areas that could be improved. The specific comments and suggestions for improvement are as follows.

  1. The wide range of topics diverts attention from the central theme of biological function. It would be advisable to streamline the relevant sections so as to keep the focus on the primary objective.
  2. For the defense hypothesis, provide more direct experimental evidence (e.g., gene knockout studies, ecological interactions) to strengthen the argument.
  3. The critique of ELISA-based studies is valid but could be balanced with a discussion of how modern mass spectrometry or genomic tools could resolve such controversies.
  4. Define acronyms (RALs, DHPLs) upon first use and avoid overuse of abbreviations to improve readability.
  5. Ensure consistent use of nomenclature (Fusarium graminearum, Gibberella zeae..).
  6. Clearer labels and a legend are needed in Figure 5.
  7. Some sections rely heavily on older literature. Update with recent studies where possible, especially for molecular mechanisms and ecological interactions.
  8. Minor grammatical errors need to be corrected (tinkering evolution should be evolutionary tinkering).

Reviewer 2 Report

Comments and Suggestions for Authors

The authors present a review entitled "Biological function of zearalenone and other resorcylic and 2
dihydroxyphenylacetic lactones (RALs and DHPLs) in fungi"

I think this review is within scope for the journal Toxins and could be published.

I have a few comments.

The abstract could be more focused on the results "The main subject
of this review is a critical assessment of the hypotheses about the biological function of ZEN" 

Also the end sentence seems out of place and copuld be incorporated in the main summary of results/findings rather than after the summary? "Similar considerations, together with recently published results, led us to suggest that 18
the biological function of the Fusarium metabolite fusaristatin A is defense against predation."

The Introduction - In general this was fine at the start but seemed to be mainly focused on the study aims/objectives. I think there could be more emphasis on summarising the main research studies related to ZEN. 

There are also partrs that read more like a summary of results/hypotheses based on the published literature "Fungi produce numerous macrolactones of resorcylic acid (RALs) and dihydroxy- phenylacetic acid (DHPLs) with structures and biological activities similar to ZEN. We hypothesize that the fundamental function of these metabolites, as well as ZEN, is protection of their producers from mycoparasites and competitors. In other words, we suggest that they are agents of chemical defense and interference competition. The hypothesis is supported by antifungal effects of these metabolites, widespread occurrence of specific lactonases in fungi, considerations about life style of saprotrophs and necrotrophs, and properties of molecular targets of RALs and DHPLs."

I feel that the history of ZEN etc. although interesting may not be necessary for this review which is specifcally focused on the Biological Function. I suppose you could argue it gives context but it seems to be overly long.

So I would say the whole of section 2 could be condensed into a slightly longer background and introduction section. The parts in the current introduction which relate to hypothesis etc. could be moved to later sections.

Section 4 outlining the production of ZEN has the feeling of a list in places. There are also statements such as this "On the contrary, the authors of some of the early reports denying the production of ZEN by F. moniliforme described their methods meticulously, and they examined many strains using several methods, see the entry for the year 1989 and the second entry for the year 1997 in Table 1." which don't really aid understanding. This whole section could be overhauled to make it much more clear as to the production of ZEN and the timelines of the research.

Table 1 is not clear in places in terms of what has been established. 

Some sub-sections are written  in quite a conversational style rather than objective scientific style e.g. "Puzzling discovery of genes for ZEN biosynthesis in F. verticillioides genome"

"To all surprise, both PKS genes involved in ZEN synthesis have been found in the genome of a particular strain F. verticillioides according a paper recently published in Pathogens [148]"

This continues with the section "In remembrance of a legend"
We finish this section by quoting W. Marasas, the doyen of research on Fusarium mycotoxins" Although I can see the purpose of this sub-section I think that it could be written more scientifically and objectively in places.

I think the sections on the various hypotheses are better structured and scienmtifically written.

Section 9 seems to be almost a separate subject matter although related to ZEN. But it does explain why the abstract treated it in that way. I think it should be better integrated to the main paper.

I think section 10 "research challenges" has some good recommendations for future directions and provides a good summary. 

Comments on the Quality of English Language

There is quite a conversational style to the English and a lack of scientific detail and treatment in places.

Round 2

Reviewer 2 Report

Comments and Suggestions for Authors

The authors answered most of my points and made extensive changes to the manuscript. Where the authors didn't make changes they explained their rationale which was convincing. Therefore, I feel that the current manuscript could be published in toxins. It is an interesting review and brings forward some new ideas.